# Small-molecule binding and sensing with a designed protein family

Gyu Rie Lee[1,2,3,4,11], Samuel J. Pellock [1,2,11], Christoffer Norn[1,2,11],
Doug Tischer [1,2], Justas Dauparas [1,2], Ivan Anishchenko [1,2],
Jaron A. M. Mercer[5,6,7,8], Alex Kang [1,2], Asim K. Bera [1,2], Hannah Nguyen [1,2],
Evans Brackenbrough[1,2], Banumathi Sankaran [9], Inna Goreshnik[1,2],
Dionne Vafeados [1,2], Nicole Roullier [1,2], Hannah L. Han [1,2],
Brian Coventry [1,2,3], Hugh K. Haddox[1,2], David R. Liu [5,6,7],
Andy Hsien-Wei Yeh [10] ✉ & David Baker [1,2,3] ✉

The de novo design of small-molecule–binding proteins holds great promise as a potential tool to develop sensors on-demand for arbitrary small molecules. Here we combine deep learning and physics-based methods to generate a family of proteins with diverse and designable pocket geometries, which we employ to computationally design binders for six small-molecule targets. Biophysical characterization of the designed binders reveals nanomolar to low micromolar binding affinities and atomic-level design accuracy. Additionally, we use a cortisol binder to design a chemically induced dimerization (CID) system that enables the construction of a biosensor for cortisol detection. The approach described here demonstrates the potential of the NTF2 fold and deep learning-based protein design in sensor development, paving the way for future platforms to design binders and sensors for small molecules across analytical, environmental, and biomedical applications.

The design of small-molecule–binding proteins with high affinity and specificity is of considerable interest. For example, biosensors and switches that undergo conformational changes upon ligand binding are broadly useful, but most approaches to develop new inducible systems focus on the discovery and engineering of natural proteins[1–3]. Previous efforts to design small-molecule–binding proteins have employed physics-based methods using naturally occurring protein scaffolds[2–6]. De novo design of 4-helix bundles and barrel-like folds have also been used as scaffolds for binding multiple cofactors and drug molecules[7–13], and algorithms to diversify the backbones for

specific folds have been developed[14,15]. However, broadly applicable approaches to diversify privileged folds for ligand binding and convert them into sensors remain limited.

Here, we show that by combining advances in deep learning-based protein structure generation and sequence design enables the leveraging of the NTF2-like fold to design a diverse set of small-molecule binding proteins that can be transformed into CID-like small-molecule sensors. This strategy builds on the premise that a large set of scaffolds housing stable pockets can support the design of binding sites for a range of small molecules, and that optimal folds must be

[1]Department of Biochemistry, University of Washington, Seattle, WA, USA. [2]Institute for Protein Design, University of Washington, Seattle, WA, USA. [3]Howard Hughes Medical Institute, University of Washington, Seattle, WA, USA. [4]Department of Biological Sciences, Korea Advanced Institute of Science and Technology, Daejeon, Republic of Korea. [5]Merkin Institute of Transformative Technologies in Healthcare, Broad Institute of Harvard and MIT, Cambridge, MA, USA. [6]Department of Chemistry and Chemical Biology, Harvard University, Cambridge, MA, USA. [7]Howard Hughes Medical Institute, Harvard University, Cambridge, MA, USA. [8]Department of Chemistry and Biochemistry, University of California Santa Cruz, Santa Cruz, CA, USA. [9]Berkeley Center for Structural Biology, Molecular Biophysics and Integrated Bioimaging, Lawrence Berkeley Laboratory, 1 Cyclotron Road, Berkeley, CA, USA. [10]Department of Biomolecular Engineering, University of California Santa Cruz, Santa Cruz, CA, USA. [11]These authors contributed equally: Gyu Rie Lee, Samuel J. Pellock, Christoffer Norn. ✉e-mail: hsyeh@ucsc.edu; dabaker@uw.edu

both compact (to keep the designs small and modular) and diversifiable (to enable generation of a wide variety of binding sites). For downstream sensing by chemically induced dimerization (CID), we seek a structural solution with the bound ligand sufficiently exposed to enable modulation of a designed dimeric protein interaction by ligand binding. Based on these criteria, we first diversify NTF2-like structures with deep-learning-based methods and install de novo protein-ligand interfaces in the globular pockets. For sequence design, we reason that the recently developed LigandMPNN, a deep learning model for protein sequence design trained on protein-small molecule complexes, can generate protein-ligand interactions more effectively than previous approaches that struggle to balance protein-ligand and intramolecular protein-protein interactions simultaneously[16].

## Results

### Computational design of small-molecule–binding proteins

The NTF2 fold is composed of 3 helices and a curved, 6-stranded β-sheet, which together form the large internal pocket characteristic of this fold family (Fig. 1A). The natural diversity of this fold is mediated by long irregular loops and unique quaternary structures, both of which affect pocket geometry and function[17,18]. We set out to design a family of NTF2s with diverse pocket geometries to accommodate a wide range of small molecules and that have minimal loops to maintain their modularity and designability. To achieve this, we generated NTF2s with family-wide hallucination[19] (Set 1: 1,615 scaffolds), redesigned these backbones with ProteinMPNN[20] and selected those that fold to the designed structure with AlphaFold[21] (Set 2: 3230 backbones), and in addition parametrically generated backbones with Rosetta[14], redesigned these with ProteinMPNN, and validated their structure with AlphaFold (Set 3: 6838 backbones) (Supplementary Fig. 1).

With a designed family of over 10,000 NTF2s with diverse pocket geometries in hand, we used RIFdock[7] to place six chemically and structurally distinct small-molecules in the central pocket of these backbones (Fig. 1B), including the stress hormone cortisol (HCY)[22], the anticoagulant warfarin (WRF)[23], the muscle relaxant rocuronium (ROC)[24], the anticoagulant apixaban (APX)[25], the antineoplastic agent SN-38 derived from the anticancer drug irinotecan (IRI)[26], and the hormone 17-α-hydroxyprogesterone (OHP)[27] (Fig. 2A). Designing polar interfaces remains an outstanding challenge in protein design[28,29], especially for small-molecule binding proteins, where an internal pocket with polar residues is needed to interact with polar functional groups of a ligand without destabilizing the protein fold. By explicitly docking polar functional groups of small molecules to pre-installed hydrogen bond networks (HBNets) present in Set 1 scaffolds (approach 1) or leveraging deep learning-based design methods trained on protein-small molecule interactions (approach 2), we reasoned that we could generate highly preorganized polar contacts while maintaining the integrity of the protein fold. For approach 1, we docked HCY, WRF, ROC, APX, and IRI into Set 1 backbones and required that at least one protein–small-molecule interaction was mediated by an HBNet residue, after which we used native-sequence guided Rosetta design[19]. Position-specific score matrices derived from NTF2 family proteins bias the sequence design. For approach 2, we used RIFdock without constraints to place OHP, APX, and IRI into scaffold Sets 2 and 3 and performed sequence design with LigandMPNN, a version of ProteinMPNN trained on small molecule-protein complexes that explicitly considers the ligand during sequence design. To select amongst the resulting designs from both approaches, we used Rosetta to calculate the number of hydrogen bonds between the protein and ligand, the binding energy (ddG), and contact molecular surface (CMS)[30] (Supplementary Fig. 2A). For design approach 2, we also selected designs based on recapitulation of the fold and binding site by single-sequence AlphaFold predictions (Supplementary Fig. 2B). After filtering, we obtained oligonucleotides encoding the binders for experimental

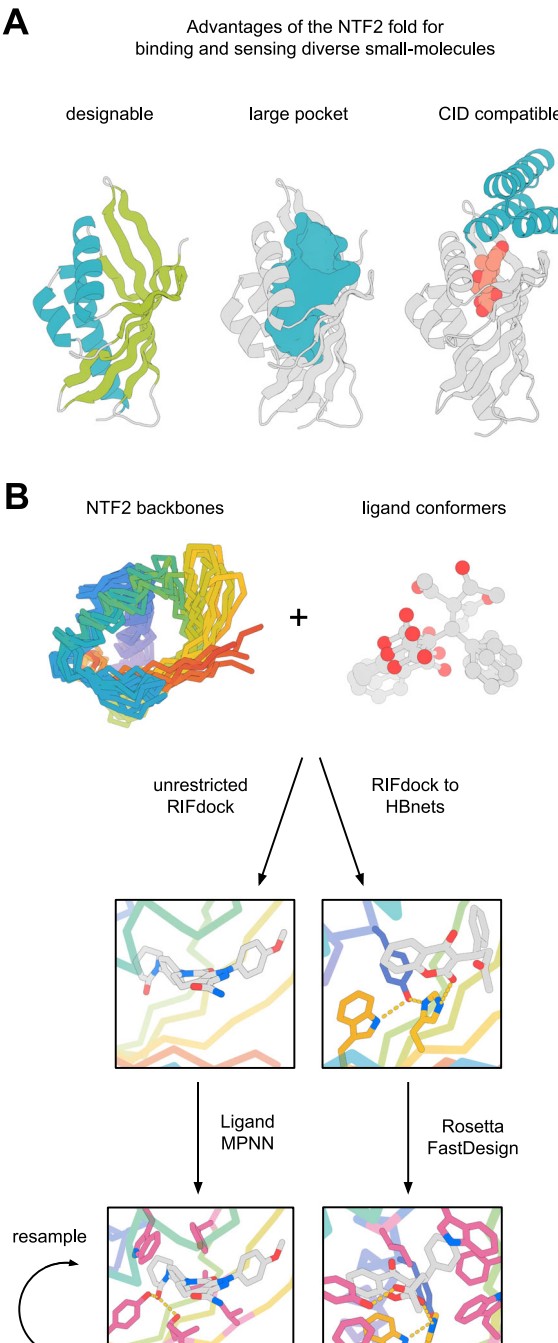

**Fig. 1 | NTF2 based design strategy for binding and sensing small molecules. A** The NTF2 fold has a designable fold made up of ideal secondary structure elements (left panel, helices in teal and strands in green), the large pocket (teal, middle panel) of NTF2s can bind various small-molecules, and the binding mode of small-molecules in NTF2s enables the generation of CIDs (right panel, NTF2 in gray, CID agent in teal, and small-molecule in red). **B** Design pipeline for small-molecule–binding in the NTF2 fold.

characterization, including 630 for HCY, 1661 for ROC, 16,276 for WRF, 9024 for APX, 19,390 for IRI, and 7573 for OHP (two approaches combined when applied).

### Characterization of small-molecule binding proteins

Designs were ordered as synthetic oligonucleotides and transformed into yeast. Binding screens were performed with yeast surface display

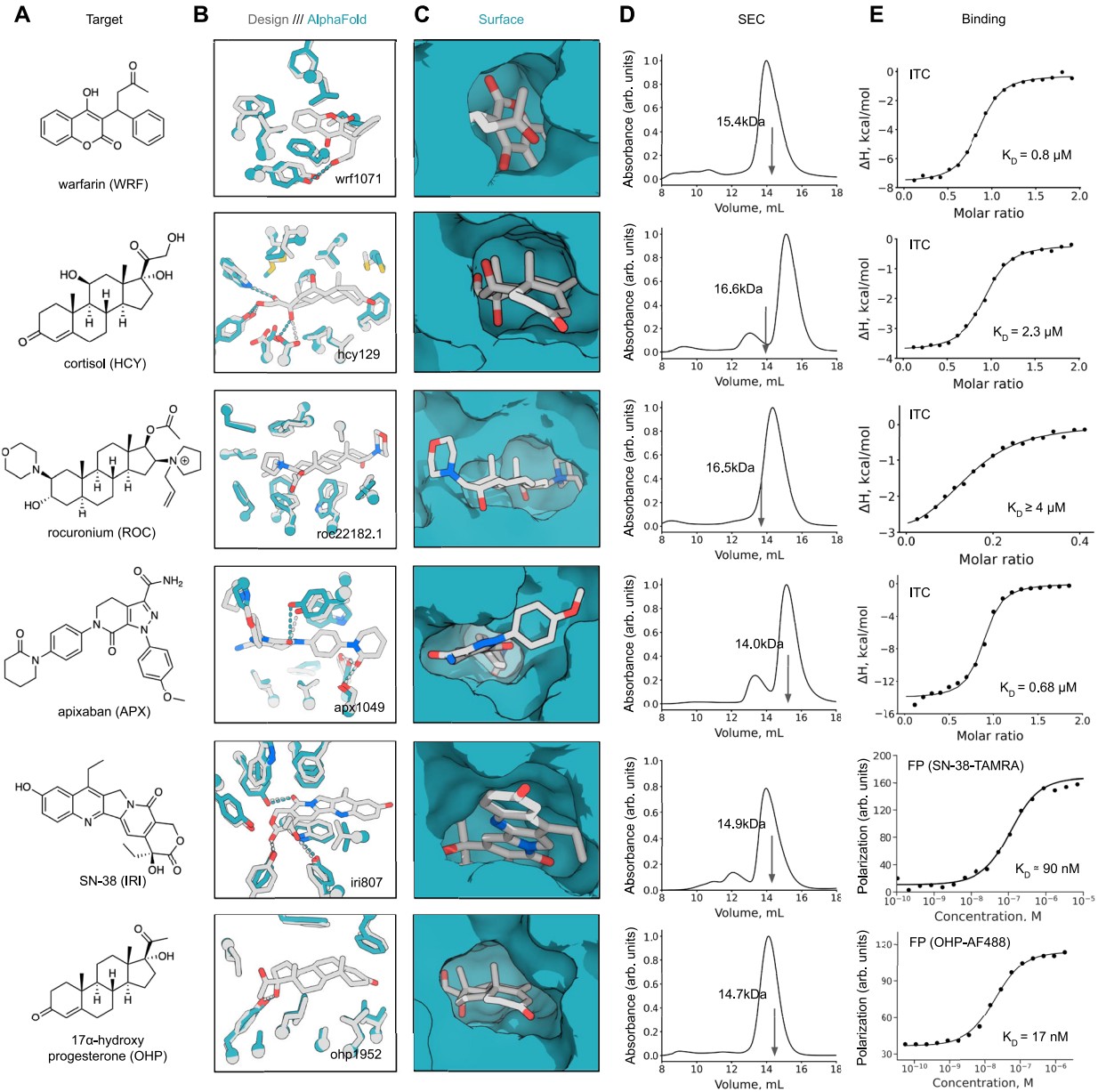

**Fig. 2 | Characterization of small-molecule–binding proteins. A** Chemical structures of small-molecule targets. **B** Overlay of design (gray) and AlphaFold (teal) models with ligands shown in gray with design name labels. **C** Surface representation (teal) of (**D**) Size-exclusion chromatography traces of select binders. The expected molecular weights of the designs are labeled in each panel. Expected elution volumes, based on the standard protein SEC profile (see Supplementary Fig. 20), are indicated with arrows (**E**) Binding analysis of small-molecule-protein interactions with ITC or FP (Table S2). Binding affinity characterized with FP used the target molecules modified with a fluorophore linker (SN-38-TAMRA and OHP-AF488). Source data are provided as a Source Data file.

coupled to fluorescence-activated cell sorting (FACS), where interaction with a biotinylated small-molecule target enables labeling with streptavidin-phycoerythrin (SAPE). Binding signal was detected for all six targets and significant enrichment was observed after multiple rounds of sorting (Supplementary Fig. 3). Deep sequencing of the final sorted populations revealed 1, 46, 19, 8, 8, and 117 unique hits for HCY, WRF, ROC, APX, IRI, and OHP, respectively. AF2 metrics for the identified hits revealed high confidence (pLDDTs ranging from 86.0 to 95.0) and accuracy (Cα RMSDs less than 2.0 Å), and Rosetta metrics indicated high chemical (target median ddG <−30) and shape complementarity (target median CMS >240) (Supplementary Fig. 4A,B). Inspection of the design models of the identified hits confirmed numerous hydrogen bonding and hydrophobic contacts to the target

ligands as well as high shape complementarity, and comparison to the AF2 models suggests atomic-level accuracy of the design method (Fig. 2A,B).

Putative binders with higher binding enrichment factors from yeast display binding screening (1, 7, 5, 5, 7, and 8 for HCY, WRF, ROC, APX, IRI, and OHP, respectively) were expressed in *E. coli* and purified to assess their solubility, oligomeric state, and binding affinity to their cognate targets. All selected designs displayed some level of expression and 23 of 33 showed distinct monodisperse peaks by size-exclusion chromatography (SEC) (Fig. 2D, Supplementary Fig. 5). The binding affinities of the purified proteins for their target small-molecules (1, 2, 1, 4, 2, and 4 for HCY, WRF, ROC, APX, IRI, and OHP, respectively) were determined by isothermal titration calorimetry

(ITC), fluorescence polarization (FP), and bio-layer interferometry (BLI), and the $K_D$ of the highest affinity binders across all targets ranged from low micromolar to high nanomolar (Fig. 2E and Supplementary Table 2). Ligands used for binding assays with FP and BLI were functionalized with fluorophores or biotin, with details provided in Supplementary Table 2. Among the characterized binders, designs that showed SEC profiles with a monodisperse peak eluting near the expected volume for the monomeric molecular weight (based on the standard protein SEC profile in Supplementary Fig. 20) tended to exhibit higher binding affinities such as apx1049, hcy129, wrf1071, ohp115, and ohp1569. Binders such as apx1501 displayed SEC profiles indicative of species larger than the expected monomeric molecular weight, which may have resulted in non-1:1 binding stoichiometry (Supplementary Fig. 15) or reduced binding affinity due to oligomerization or altered folding states.

Targets HCY and OHP are chemically similar to steroid-like ligands known to bind native ketosteroid isomerases (KSIs) with NTF2-fold structures. In retrospect, we designed binders with higher binding affinities towards these steroid-like targets compared to less chemically similar targets, such as ROC and IRI (Supplementary Tables 2, 5). Despite APX exhibiting low chemical similarity (Tanimoto coefficient <0.2) to the KSI-bound ligands in the PDB, we could design a binder achieving nanomolar-range binding affinity (apx1049). Nevertheless, the success rate of high-affinity binder designs in our approach was lower than alternative methods utilizing the four-helical bundles[9], and this highlights the need to develop methods capable of designing scaffolds to accommodate a broader range of targets[31] and accurately predicting binding prior to experimental validation.

A retrospective analysis revealed that the Set1 and Set2 NTF2 scaffolds exhibited higher structurally similarity to the NTF2-fold ketosteroid isomerase (KSI) family compared to the Set3 NTF2 scaffolds (Supplementary Figs. 1, 9). The characterized binders were derived from Set1 or Set2 backbones and showed structural similarity to the KSI proteins complexed to steroid-like ligands (Supplementary Table 4). These observations suggest that, despite our computational exploration of a broader conformational space within the NTF2-like fold, the backbone structures leading to functional designs closely resembled native KSI structures.

We performed site saturation mutagenesis (SSM) experiments on a selected set of binders (1, 26, 1, 1, and 2 for HCY, WRF, ROC, APX, and IRI, respectively) identified from yeast display screening. The binders were selected based on binding affinity based on the previous characterizations to assess the sequence-function relationship and confirm the designed binding mode (Supplementary Fig. 6). For WRF, we pooled the SSM libraries of 26 potential hits from the yeast binding screening for further selection, and this led to selecting the 7 binders, which were further expressed in *E. coli* and examined after protein purification (Supplementary Fig. 5). Overall, analysis of the SSMs revealed high conservation of the designed protein-ligand interactions, including key hydrogen bond and hydrophobic interactions (Supplementary Fig. 6A). Notably, protein-ligand contacts from pre-installed HBNets, such as H98 in wrf7190 and Q101 and Q117 in roc22182 were conserved in the SSM (Supplementary Fig. 6C). Hydrogen bonding interactions designed by approach 2 based on LigandMPNN (S95 and Y108 in apx1049, and Y35, Y53, and Y82 in iri807) were also highly conserved (Supplementary Fig. 6C). Furthermore, key residues that preorganize these residues were also conserved (H98, Y14, W100 for wrf7190, W14, Y30, Q117 for roc22182, and W18, S95, Y108 for apx1049). Both design approaches generated SSM-conserved pi-pi or CH-pi interactions (approach 1: F37 in wrf1071, approach 2: W18 and Y61 in apx1049 and F64 in iri807, Supplementary Fig. 6C).

For rocuronium, the most enriched hit identified by yeast display showed poor folding by SEC (Supplementary Fig. 5), so we optimized this sequence by combining favorable mutants from the SSM and a mutation based on an AlphaFold multimer prediction that showed the formation of a dimer that likely precludes binding to rocuronium (Supplementary Figs. 6, 7). Combining these mutations improved solubility and enabled characterization of binding (roc22182.1: Y30W/Y50V/F36L/I38W/W56E/F61Q/V88L/Q97D/N119F, Fig. 2). Despite our efforts to reduce aggregation via mutagenesis and redesign, the binding stoichiometry observed for roc22182.1 was very low (approximately 1:10). This suggests that binding behavior remains suboptimal, possibly due to residual aggregation of the protein or ligand, or cooperativity effects. ROC-binding proteins were designed exclusively using approach 1 (NTF2 scaffold Set1) in this work, and we anticipate that applying alternative strategies such as design approach 2 or employing different scaffolds will yield binders with improved performance. We also attempted to optimize the most highly enriched hit for SN-38 (iri807) by incorporating mutations identified in an SSM experiment, and the $K_D$ improvements of selected variants ranged from 1.5 to 5.5-fold based on FP using SN-38-TAMRA (Supplementary Table 2).

### Structural characterization of designed binders

To further assess the accuracy of designed small-molecule binding proteins, experimental structures of protein-ligand complexes were determined. We successfully obtained crystal structures of the cortisol binding protein hcy129 (design approach 1) and apixaban binding protein apx1049 (design approach 2). To aid in crystallization of the cortisol binder hcy129, ProteinMPNN was used to redesign the surface given its previous success in yielding crystallizable sequences[32]. We crystallized the ProteinMPNN-redesigned version of hcy129, hcy129_mpnn5, and obtained the 1.5 Å crystal structure of this design in complex with cortisol (Fig. 3A and Supplementary Fig. 8). Structural alignment of the crystal structure to the design model revealed a Cα RMSD of 1.1 Å over 120 residues (Fig. 3A). In addition to the overall accuracy of the fold, key hydrogen bonding residues, such as W17, Y18, and Y108, and the binding orientation of cortisol closely match the design model (Fig. 3B). Notably, multiple polar interactions are mediated by a pre-installed HBNet unique to scaffold Set 1, demonstrating that this approach worked as intended to generate preorganized polar contacts.

The apixaban binder apx1049 was crystallized in complex with apixaban and a 2.1 Å resolution crystal structure was determined. The structure shows sub-angstrom agreement to the design model with Cα RMSD of 0.6 Å over 113 residues (Fig. 3C). The designed protein-ligand interactions were quite accurate, with the crystal structure nearly identical to the design model for key hydrogen bonds from residues S95 and Y108 and pi-pi stacking with aromatic residues W108 and Y61 (Fig. 3D) that lock the ligand in the targeted conformation by orienting the central apixaban ring. Taken together, the contacts mediated by apx1049 to apixaban form an extensive set of contacts that form a highly shape complementary pocket (Fig. 3D).

### Specificity assessment of designed binders

We conducted binding assays for each of the six representative binders against all six target ligands to evaluate their specificity profiles. The selected binders used for the evaluation and their binding affinities ($K_D$) are detailed in Supplementary Table 6 and Supplementary Fig. 10. We also measured binding of the six target ligands with albumin, a protein known for its promiscuous binding to hydrophobic ligands (Supplementary Table 6). Binders such as hcy129.1, iri807.1 and apx1049, which demonstrated high binding affinities toward their cognate targets, showed higher target specificity over other ligands. Notably, ohp1952, which was designed to bind 17α-hydroxyprogesterone (OHP), exhibited tight binding to the ligand cortisol (HCY) as well, indicating low target specificity. The designed binders showed preferential binding to their cognate targets, whereas albumin exhibited no binding to the target ligands HCY, IRI, APX, and

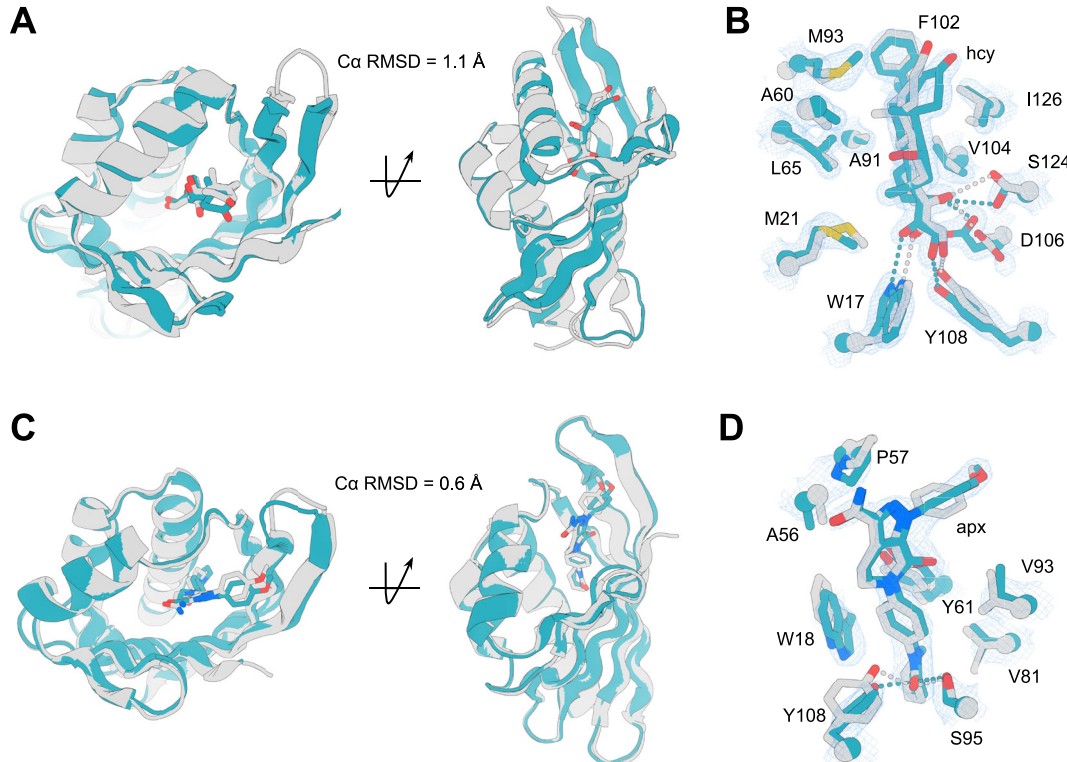

**Fig. 3 | Structural analysis of designed cortisol and apixaban-binding proteins.** **A** Structural superposition of design model (gray) and crystal structure (teal) of hcy129_mpnn5. **B** Zoom-in of binding pocket of hcy129_mpnn5 (gray) overlaid on the crystal structure (teal) with 2Fo-Fc map shown for cortisol at 1.5 σ. **C** Overlay of design model (gray) and crystal structure (teal) of apx1049. **D** Zoom-in of the binding pocket of apx1049 (gray) overlaid on the crystal structure (teal) with 2Fo-Fc map shown for apixaban at 1.5 σ.

ROC, and weak binding to OHP. However, for warfarin (WRF), albumin exhibited a micromolar-range binding affinity to WRF-TAMRA ($K_D$ 5.0 μM), which is in a similar range to that of the binder wrf1071 ($K_D$ 1.1 μM). These results indicate that further improvements in our design approach are necessary to enhance specificity and sensitivity, especially regarding targets with chemical similarities (such as HCY and OHP), and hydrophobic ligands (such as WRF).

**Design and characterization of a cortisol-induced heterodimer**
Cortisol is typically present at low nanomolar concentrations in physiological samples, and diseases such as Cushing's syndrome can be diagnosed when the plasma cortisol level is higher than 38 nM[33,34]. To improve the binding affinity of hcy129 for use as a cortisol biosensor, we screened a library of combinatorial mutants based on favorable mutations identified from the SSM of hcy129 (Fig. 4A, B) by yeast display, and observed significant improvements in binding affinity (Supplementary Fig. 11). We selected the best variant from this library for production in *E. coli* and characterization by ITC. This variant, hcy129.1, displayed a $K_D$ of 68 nM, a 31-fold improvement in affinity compared to the original design (Fig. 4C). Docking cortisol into the AlphaFold model of hcy129.1 with GALigandDock[35] revealed that improvements in affinity are likely a result of improved hydrophobic interactions with cortisol (Fig. 4D).

With a binder that recognizes cortisol at physiologically relevant concentrations in hand, we set out to design a cortisol-dependent heterodimerization system. The binding mode of cortisol in the hcy129.1 leaves part of its structure exposed, enabling the design of proteins that form a ternary complex that interfaces with the ligand bound state of hcy129.1 (Fig. 4E). Next, the surface at the opening of the pocket of hcy129.1 was redesigned to facilitate docking and design

of protein binders to the complex (see **Methods**). After generating this new variant, hcy129.1_CID, we performed RIFdock against the hcy129.1_CID-cortisol complex with a miniprotein scaffold library[30]. This scaffold library was designed to form more stable cores and has previously proven successful for designing protein-protein binders. This approach generated numerous ternary complexes where both the miniprotein and hcy129.1 interacted with each other and also to cortisol (Fig. 4E). Sequence design of the resulting docks was carried out with both Rosetta FastDesign and ProteinMPNN, and the heterodimeric complexes were computationally selected with AlphaFold2 structure prediction applying cutoffs of pAE <10 and plDDT >85.

The designed minibinders were displayed on yeast and the binding to biotinylated hcy129.1_CID was assessed in the presence or absence of cortisol by FACS (Supplementary Fig. 12A). Populations enriched for binding to hcy129.1_CID in the presence of cortisol were collected and colonies were picked to identify the mini binder, miniH11, via FACS analysis (Supplementary Fig. 12B). We expressed, purified, and combined both hcy129.1_CID and miniH11 in the presence or absence of cortisol and analyzed them by SEC in which we observed a shift towards a higher molecular weight species in the presence of cortisol (Fig. 4F). We also analyzed the cortisol-induced complex by native mass spectrometry[36], which revealed a molecular weight for the hcy129.1_CID-cortisol-miniH11 ternary complex (Supplementary Fig. 13A). Together, these data suggest that the designed proteins form a cortisol-dependent heterodimerization.

As a proof-of-concept for cortisol sensing based on the designed CID, we genetically fused hcy129.1_CID and miniH11 to the SmBiT and LgBiT components of the NanoBiT system, respectively, which reconstitutes the NanoBiT luciferase and generates luminescence when brought in close proximity by a molecular interaction[37] (Fig. 4G).

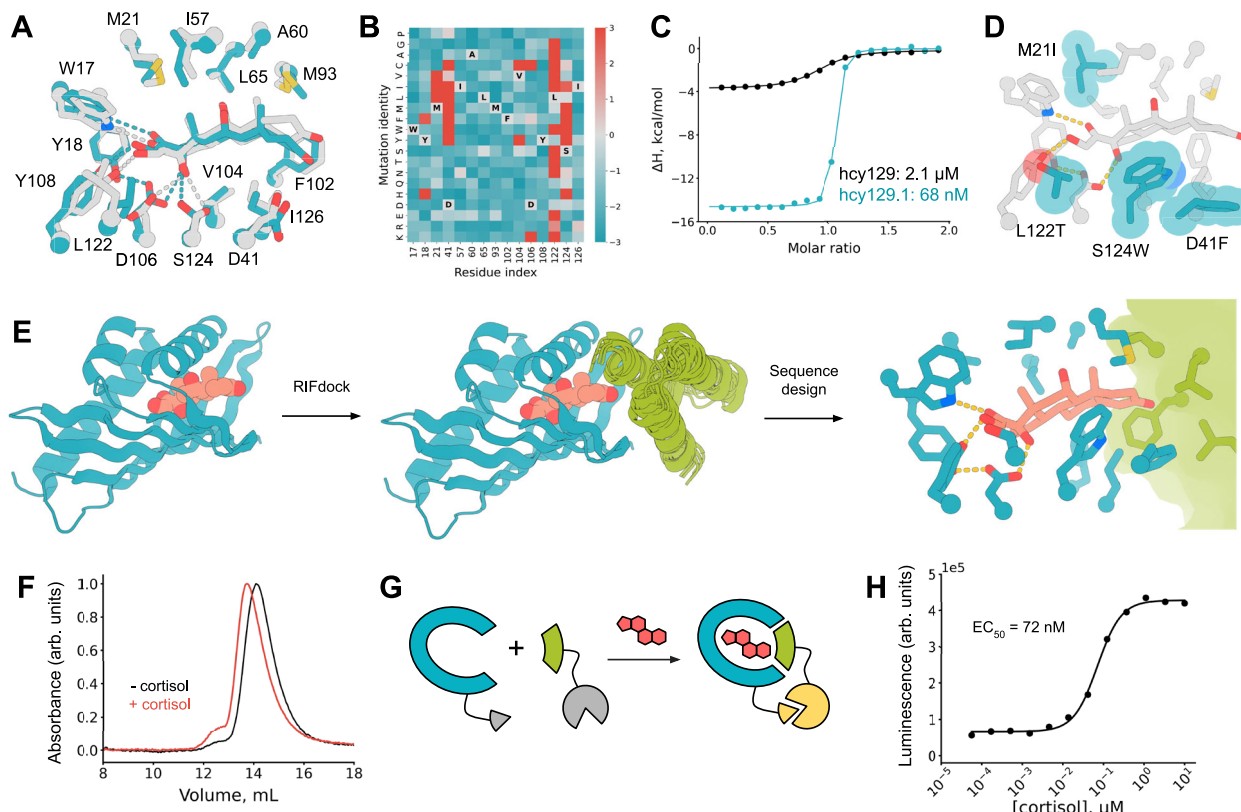

**Fig. 4 | Design and characterization of a chemically-induced heterodimer for cortisol sensing. A** Zoom-in of the binding site of hcy129_mpnn5, design model (gray) and crystal structure (teal) overlayed. **B** Heatmap showing enrichment of each mutation compared to wild type from the site saturation mutagenesis experiment (teal: weaker binding, gray: no change, red: tighter binding). **C** ITC binding isotherms of hcy129 and hcy129.1. **D** Zoom-in of the binding site of an AlphaFold-predicted structure of hcy129.1 with cortisol docked using GALigandDock. Mutated residues show in teal. **E** Design pipeline for cortisol-induced heterodimerization. Starting from a model of triple mutant cortisol binder hcy129.1_CID (teal, R43I/R95Q/Q128L) in complex with cortisol (peach), we used RIFdock and a library of designed 3-helix bundles (green) to generate thousands of putative ternary complexes (middle), and amino acid sequences encoding these complexes were generated with Rosetta FastDesign and ProteinMPNN. Zoom-in of the designed CID ternary complex (hcy129.1_CID in teal, miniH11 in green, and cortisol in peach). **F** Size-exclusion chromatography traces of an equimolar solution of hcy129.1_CID and miniH11 (1 μM) in the presence (red) or absence (black) of cortisol (10 μM). **G** Schematic diagram of the designed cortisol-induced heterodimerization coupled with a binary split luciferase to create a biosensor. **H** Cortisol-dependent luminescent response in an equimolar solution of hcy129.1_CID-SmBiT and miniH11_LgBiT (200 nM). Source data (**C,H**) are provided as a Source Data file.

We expressed and purified the CID-NanoBiT fusion constructs and titrated them with cortisol, which generated a luminescent signal with an estimated $EC_{50}$ of 72 nM (Fig. 4H). This closely matches the $K_D$ of hcy129.1 for cortisol, which suggests that a specific interaction between hcy129.1_CID and cortisol promotes association of hcy129.1_CID and miniH11. To assess the affinity of the CID components in the absence of cortisol, we titrated miniH11-LgBiT with increasing concentrations of hcy129.1_CID-SmBiT, which revealed an estimated $K_D$ of >5 μM (Supplementary Fig. 13B), at least 2 orders of magnitude weaker than the $EC_{50}$ identified for cortisol-induced dimerization, indicating that the dimerization observed at low concentrations of the CID components is dependent on the presence of cortisol. Taken together, these data demonstrate that the NTF2-based small-molecule binders designed in this study can be engineered to serve as biosensors.

The selectivity of the cortisol sensor was evaluated against structurally similar ligands: cortisol, cortisone, corticosterone, 17α-hydroxyprogesterone (OHP), warfarin, and rocuronium (Supplementary Fig. 14). The sensor responded to cortisol, cortisone, and OHP, but not to corticosterone. This lack of response is likely due to the absence of a hydroxyl group at the 17-position carbon in corticosterone, which is essential for a key polar interaction in our design model (Fig. 4D). Cortisol (-OH), cortisone (C = O), and OHP (H) differ only at the 11-position carbon—a functional group we did not specifically design an

interaction for. Overall, the selectivity profile is consistent with our model, though further redesign and method development will be required to enhance the designed sensors' sensitivity and specificity.

## Discussion

By integrating deep learning-based methods for backbone generation, ProteinMPNN and LigandMPNN for sequence design, and AlphaFold for structural filtering, we show that scaffold sets inspired by a privileged ligand-binding fold and deeply sampled in both sequence and structural space can be utilized to design functional proteins. This was exemplified by the design of binders to six different small molecules. Furthermore, we showcased the potential of NTF2 fold-based binders as a foundation for the design of chemically induced dimerization (CID) systems in biosensing applications. Notably, many of the high-affinity binders we generated targeted ligands with chemical similarity to those recognized by the NTF2-fold ketosteroid isomerase family. Looking ahead, the generality of our approach may be further improved by recent advances in deep learning-based structure generation tailored to individual target compounds, enabling the design of binders for a broader spectrum of small-molecules. Additionally, improving target specificity remains an important direction for ongoing and future research. We envision that more accurate prediction of protein–ligand interactions, combined with design methodology developments that can leverage unique protein-ligand interaction

patterns such as target-specific hydrogen-bonding, will advance the design of highly specific small-molecule binders and sensors.

## Methods

### Statistics & Reproducibility

No statistical method was used to predetermine sample size and no data were excluded from the analyses. The experiments were not randomized.

### Generation of NTF2 scaffolds

Set 1 backbones were generated based on family-wide hallucination approach combined with native context-biased sequence design[19]. Protein sequences were optimized by Markov chain Monte Carlo sampling with constraints derived from NTF2-like structures. For set 2 and set 3 backbones, hallucinated[19] and algorithmically generated[14] backbones were used as input for ProteinMPNN. For hallucinated backbones, two separate ProteinMPNN design approaches were taken, one in which native HBNet residues were kept fixed and one in which the entire protein was allowed to be redesigned, and both design runs were performed with and without polar residue-biased weights on amino acid (aa) compositions. The algorithmically generated backbones were also designed with and without a polar-aa bias term applied. Structures of resulting sequences were predicted by Alpha-Fold2 in single-sequence mode, using model 4 with 10 recycles, and structures with Cα RMSDs less than 1 Å compared to the original scaffold and an average plDDT greater than 92 were selected for docking.

### Generation of ligand conformers

For ROC, HCY, WRF, APX, and IRI, 2D chemical structures were made in ChemDraw and exported as SDfiles or SMILES strings, which served as input for RDKit to generate initial 3D models of the target structures with the ETKDGv2 method[38]. For the first set of designs based on Set 1 backbones, partial charges were assigned using antechamber[39] and geometry optimization was performed using xtb[40]. Additional sampling of ligand conformers was performed with the CSD conformer generator[41] and final sets of 15 conformers of fewer were selected manually as input for RIFgen. 200 conformers of OHP, APX, SN-38 were generated for each ligand in design approach 2 and the structures were optimized using the MMFF force field[42]. The conformers were clustered based on pairwise RMSD. 1, 26, and 4 cluster representatives with the lowest energy were selected respectively for OHP, APX, SN-38, and final minimization of the ligand conformations was done by using the AIMnet potential[43].

### Rosetta ligand parameter generation

Conformers generated for each target small-molecule were converted into MOL2 format using openbabel and used as inputs to generate Rosetta parameter files. These parameters were provided when using the generic potential to perform Rosetta calculations[35].

### Computational design of small-molecule-binding proteins

Backbone generation, ligand docking, and sequence design for Set 1 backbones was conducted using a combination of deep-learning-based and physics-based methods[19]. Backbones were generated with the TrRosetta-based hallucination method using constraints derived from NTF2-like structures and homology models including sequence and geometric constraints for recapitulation of native hydrogen-bonding networks[19]. These hydrogen-bonding networks (HbNet) were utilized for docking the polar functional groups of the small-molecules. For each target small-molecule, functional groups that can serve as hydrogen bond donors or acceptors were identified, and biased RIFdock was performed for all possible combinations of ligand atom and HbNet residue pairs. The bias was applied using the tuning_file flag during the Rifgen step of RIFdock, which allowed

specification of the type of interaction (such as hydrogen bonding) between a specific amino acid type and target ligand atom pair. The resulting biased RIF (Rotamer Interaction Field) was then used to dock the target ligand onto the NTF2 scaffolds (Set 1), enforcing the defined hydrogen bonding interactions. This calculation was repeated for all possible amino acid-ligand atom hydrogen bonding pairs. The sequences of the resulting NTF2-ligand docks were subsequently redesigned. Sequence design was performed using Rosetta FastDesign[44,45] with position-specific scoring matrix (PSSM) sequence constraints, while the amino acid identities of the residues forming hydrogen-bonds to the target ligand were kept fixed. The PSSM was generated from aligned NTF2-like sequences[19]. Rosetta metrics such as ddG, contact molecular surface (CMS), solvent accessible surface area between the ligand and the protein (SASA), and the number of hydrogen bonds (nHb) were used to select the designs to test after clustering based on sequence identity (Supplementary Table 1).

For design approach 2, all conformers of OHP, APX, and IRI were docked to scaffolds Set 2 and 3 using unconstrained RIFdock. Unconstrained RIFdock also employed the tuning_file flag during the rifgen stage, but it used a file listing all possible hydrogen bonding interactions between multiple amino acids and target ligand atoms, without incorporating any sequence information from the NTF2 scaffolds (Set 2). The docks were initially filtered based on their shape complementarity to the scaffold, estimated by the protein-ligand contact molecular surface area calculated using Rosetta[30]. The filtered docks underwent an initial round of sequence design using LigandMPNN followed by Rosetta FastRelax[46,47]. The backbone and the coordinates of the ligand atoms were used as input for LigandMPNN and no sequence constraints were applied at this stage to completely redesign the sequence. Eight sequences were generated with temperature T = 0.2 per input, and 3 iterations of LigandMPNN and RosettaPackMin (before the last step) or RosettaFastRelax (last step) were performed[45]. Distance restraints were applied for selected sets of ligand atoms and protein atoms in the binding pocket to keep the ligand in place after sequence design. We used ligand atoms of a central chemical group that doesn't include rotatable bonds to generate distance restraints to neighbor protein Cα atoms within 10 Å. The restraints were applied with harmonic potentials with a standard deviation of 0.5 Å. The designs were filtered using Rosetta metrics, including ddG, contact molecular surface (CMS), and the number of hydrogen bonds (nHb) formed between the protein and ligand. We used AlphaFold2 (model 4, 10 recycles) to validate the designed sequences, and filtered based on plDDT, Cα RMSD, and binding site residue side chain RMSD (Supplementary Table 1). To increase the sequence diversity while maintaining structure-prediction quality of the designed sequences, GALigandDock was performed on the AF2 models of the selected designs, and the docking solutions with ligand all-atom RMSD lower than 2 Å were used as inputs for the next iterative round of sequence design. This docking process enabled greater sampling of ligand conformers and rigid body positioning within the pocket. A second round of sequence design based on the same design scheme consisting of LigandMPNN and RosettaRelax was applied to the updated inputs, and the final set of designs to be ordered was selected using Rosetta and AF2 metrics (Supplementary Table 1).

### Library assembly and yeast display screening

DNA encoding designed small-molecule–binding proteins were ordered as single-stranded synthetic oligos from Twist Bioscience and assembled using qPCR[48]. The assembled library DNA was mixed with linearized pETCON3 plasmid and the resulting solution was transformed into competent EBY100 yeast by electroporation[49]. The yeast library was first sorted for expression, and any cells labeled by anti-c-Myc-fluorescein isothiocyanate (anti-c-Myc-FITC, Immunology Consultants Laboratory) were collected and then grown for 2-3 days in c-Trp -Ura selection media. To assess binding, cells collected from the

expression sort were incubated with biotinylated ligand, anti-c-Myc-FITC, and streptavidin conjugated to phycoerythrin (SAPE, Thermo-Fisher) in PBSF (phosphate buffered saline with 0.1% bovine serum albumin). Yeast cells labeled by both anti-c-Myc-FITC and SAPE were collected and this same process was repeated for at least two more rounds of sorting. The concentrations of the biotinylated ligand and SAPE were chosen differently for each sort (Supplementary Fig. 3). Final cell populations were prepared for deep sequencing or streaked in c -Trp -Ura plates and individual colonies were sequenced by colony PCR and Sanger sequencing to identify functional sequences.

## Site saturation mutagenesis library preparation and sorting
For each putative binder amino acid sequence, single site positions were mutated to all possible amino acids and were ordered as synthetic oligos from Twist or Agilent. Oligos with mutations on the N terminal half side were assembled with a wild type sequence gblock ordered from Integrated DNA Technologies (IDT). The oligos and the gblock were constructed to have constant overlap for assembly. The same was applied to the oligos with mutations on the C terminal half, and after combining the two DNA pools, yeast cell transformation by electroporation and FACS were performed as described in the preceding paragraph.

## Combinatorial mutagenesis library preparation and sorting
**Cortisol binder hcy129.** DNA fragments with combinatorial mutations were ordered as eblocks from IDT with BsaI restriction sites and pET-CON3 vector overlapping sequences. Eblocks corresponding to N or C terminal ends of the sequence were pooled and were stitched together using Golden Gate assembly. The assembled library was chemically transformed[50] to competent EBY100 yeast cells with the linearized pETCON3 vector.

**Rocuronium binder roc22182.** We ordered synthetic oligonucleotides including degenerate codons (opools, IDT) optimized with SwiftLib[51] to construct a combinatorial mutagenesis library. Oligonucleotides were cloned into a pETCON3 vector and transformed into yeast cells using electroporation.

**SN-38 binder iri807.** The structures of all possible combinatorial mutant sequences of iri807 were predicted using AlphaFold2, and 153 sequences that yielded predictions with pIDDT > 90.0 and Cα RMSD < 1.5 Å were chosen. The DNA fragments of the selected sequences were ordered as eblocks, chemically transformed to yeast cells, and the binding of each clone to SN-38 was analyzed using the Attune flow cytometer (Thermo Fisher). The sequences with the best binding signals were tested for binding using BLI and FP.

## Deep sequencing and analysis
The collected yeast cells after FACS were grown in c -Trp -Ura media with 2% glucose for 2 to 3 days, and 3e7 to 5e7 cells were used to extract the plasmids (Zymoprep, Zymo Research). Two rounds of qPCR amplifications were performed using the same protocol as the design library assembly for DNA amplification and attachment of Illumina and pool specific barcodes. The purified DNA samples were sequenced using Illumina MiSeq sequencing.

The sequencing outputs were downloaded in FASTQ format, and we used the program PEAR[52] to merge the paired end reads. Reads matching the ordered design amino acid sequences were counted and used for analyzing the binding enrichments. For each sequence the frequencies were calculated for all sorts, and the log ratio of the binding frequency to the expression frequency was used for analysis. For SSM sorts, the ratio of mutant counts from a binding sort to the expression sort was compared to that of the wild type sequence, and the log difference was defined as an enrichment factor[53].

## Synthesis of biotinylated small-molecules
Reactions were performed in oven-dried glassware with magnetic stir-ring. Flash column chromatography was performed using silica gel (pore size 60 Å, 230–400 mesh particle size, 40–63 μm particle size) from Sigma-Aldrich. Analytical thin-layer chromatography (TLC) was carried out on precoated silica gel plates (TLC Silica gel 60 F254, 250 μM thickness) from EMD Millipore. TLC plates were visualized with ultraviolet light or stained by submersion in a solution of p-anisaldehyde followed by heating. p-Anisaldehyde TLC stain was prepared by sequential addition of sulfuric acid (5.0 mL), glacial acetic acid (1.5 mL), and p-anisaldehyde (3.7 mL) to absolute ethanol (135 mL) at room temperature with stirring. Analytical liquid chromatography–mass spectrometry (LCMS) was performed on a Waters Acquity UPLC coupled to an SQD2 single quadrupole mass spectrometer with an electrospray ionization (ESI) source. Preparative high-performance liquid chromatography (HPLC) was performed on an Agilent 1260 Infinity coupled to an Agilent 6120 single quadrupole mass spectrometer with an electrospray ionization (ESI) source. High-resolution mass spectra were obtained via reverse-phase LC/MS on an Agilent G6230B TOF on an AdvanceBio RP-Desalting column, and subsequently deconvoluted by way of Bioconfirm using a total entropy algorithm. The structures of small-molecule ligands conjugated to biotin, reaction schemes, and characterization of the products are available in the "Synthesis of biotin conjugated small-molecules" section of the Supplementary Information.

## Expression and purification of designed small-molecule–binding proteins
DNA sequences of designed small-molecule binders were ordered as eblocks from Integrated DNA Technologies and cloned into an N-terminal hexahistidine tag-containing vector via the golden gate method using the BsaI restriction enzyme[32,54]. The assembled plasmid was transformed into E. coli BL21(DE3) and resultant transformants were cultured in autoinduction media at 37 °C for 3–4 h and then lowered to 18 °C overnight. Cells were harvested by centrifugation at 4000 × g for 10 min. Cells were resuspended in lysis buffer (40 mM imidazole, 100 mM sodium phosphate, 500 mM sodium chloride, pH 7.4 or 40 mM imidazole, 100 mM Tris-HCl, and 500 mM sodium chloride, pH 8.0) and then sonicated on ice for 5 min with 10 s on and 10 s off. Resulting lysates were centrifuged for 30 min at 14000 × g to clarify the lysate. The clarified lysates were applied to ~1 mL of nickel resin and washed with 10 column volumes of wash buffer (40 mM imidazole, 100 mM sodium phosphate, and 500 mM sodium chloride, pH 7.4 or 30 mM imidazole, 20 mM Tris-HCl, and 500 mM sodium chloride, pH 8.0). A pre-elution wash of 400 μL was performed with elution buffer (400 mM imidazole, 100 mM sodium phosphate, or 500 mM sodium chloride, pH 7.4 or 500 mM imidazole, 20 mM Tris-HCl, and 100 mM NaCl, pH 8.0). Samples were eluted with 1.3 mL of elution buffer and then filtered through a 0.2 μm filter prior to SEC purification with a superdex 75 10/300 increase column. Final purified samples were snap-frozen in liquid nitrogen and stored at −80 °C.

## Characterization of binding affinity by ITC
Binding affinity of designed proteins was assessed by titrating purified protein with ligand using a microcal PEAQ ITC-Automated instrument. Both protein and ligand were prepared in identical buffers that were degassed by bottle-top filtration. Cortisol titrations contained 1% DMSO in all solutions and warfarin and apixaban titrations contained 5% DMSO in all solutions. For rocuronium binding, no DMSO was present and titrations were performed in 20 mM HEPES, 50 mM NaCl, pH 8 buffer. All titrations were performed at 25 °C and 19 total injections, 0.4 μL for the first injection and 2 μL for the remaining 18. Resulting titration data were analyzed with the Malvern Panalytical ITC Analysis software, and a single-site binding model was used to fit the resulting data.

## Characterization of binding affinity by BLI

Biolayer interferometry (BLI) was used to estimate the binding affinity and screen for binding of the purified proteins and target small-molecules APX, and SN-38. HBS-EP+ buffer (Cytiva) with 0.1% Bovine Serum Albumin and 0.01% Tween20 was used as the assay buffer and to prepare the purified protein and functionalized ligand solutions. 1 µM of small-molecule targets with biotin conjugated (APX-Biotin and SN-38-Biotin) were used as ligands to be immobilized to the Octet Streptavidin biosensors (Sartorius). Titration experiments were performed at 25 °C, and the kinetic response data was collected using Octet RED96 and R8 (ForteBio). The resulting responses of the analytes were corrected using the signal from the reference buffer, and further analysis was performed using the Octet Data Analysis software.

## Characterization of binding affinity by FP

Fluorescence polarization was performed to determine the binding affinity of the OHP binder designs. The concentration of the fluorophore labeled version of 17α-hydroxyprogesterone and SN-38, OHP-AF488, OHP-TAMRA and SN-38-TAMRA, were kept constant at 6 nM in phosphate buffered saline (PBS, Fisher BioReagents). Two-fold serial dilutions of the binder protein were prepared in 24 wells with constant amounts of the target ligand, and the plate was incubated at room temperature, shaking for at least 30 min. Fluorescence polarization filter cube (Red FP, EX 530/25 nm;EM 590/35 nm, Agilent) was used to measure polarization with the plate reader (Synergy Neo2, Biotek). A binding isotherm model was used to fit the data and estimate $K_D$.

## Surface redesign of hcy129 with ProteinMPNN

The design model of hcy129 was used as input to ProteinMPNN, and sequence design was performed with a temperature of 0.1 and cysteine was excluded during design. 10 sequences were generated and single-sequence structure predictions with AlphaFold were used to confirm that surface redesigns matched the original design model. Designs with Cα RMSDs less than 1 Å and plDDTs greater than 90 were manually inspected and designs that recapitulated the binding residue rotamers were selected for gene synthesis, biochemical characterization, and crystallography.

## Crystallography and structure determination of hcy129_mpnn5 bound to cortisol

For crystallography, hcy129_mpnn5 was expressed at 0.5 L volume in 2.8 L flasks in autoinduction media and incubated in a shaking flask at 250 rpm for 4–6 h at 37 °C and then lowered to 18 °C and incubated overnight. Purification was performed as described above, except that the elution buffer for SEC was 100 mM CHES, 100 mM acetone oxime, 100 mM NaCl, pH 8.6. To the SEC eluate ([protein] ~ 1 mg/mL), 0.5 M guanidinium hydrochloride was added and incubated for 10 min, after which $NiCl_2$ was added to a final concentration of 1 mM and incubated overnight at room temperature for sequence-specific nickel-assisted tag cleavage (SNAC)[55]. After SNAC tag cleavage, the protein was applied to a nickel affinity column to remove the free tag as well as any uncleaved protein. Flow-through as well as 3–5 mL of wash buffer eluate were collected from the nickel column. A final SEC purification was performed in 20 mM HEPES, 50 mM NaCl pH 7.4 and the eluate was concentrated to 10 mg/ml and incubated with 1 mM cortisol prior to crystallization. This combined solution was used for crystal screening using a Mosquito LCP by STP Labtech. Crystals grew in 0.2 M ammonium sulfate, 30% w/v PEG 8000 and were harvested directly from a screening tray, cryoprotected with 25% ethylene glycol, and stored in liquid nitrogen. X-ray diffraction was performed at APS 24ID-C, data were processed with XDS[56], and the structure was phased by molecular replacement using the designed structure as the search model and refined with Phenix[57]. Data collection and refinement statistics are recorded in Supplementary Table 3. Data deposition, atomic coordinates, and structure factors reported for the protein in this paper have been deposited in the Protein Data Bank (PDB), http://www.rcsb.org/ with accession code 8UQF.

## Crystallography and structure determination of apx1049 bound to apixaban

The apixaban binder apx1049 with a TEV protease cleavage site present after the N terminal hexa-histidine tag was expressed at 0.5 L volume in 2.8 L flasks in autoinduction media and incubated in a shaking flask at 250 rpm overnight at 37 °C. Protein purification was performed using nickel affinity columns and 500 mM imidazole, 20 mM Tris-HCl, 100 mM NaCl, pH 8.0 buffer as the elution buffer. The eluate was incubated with TEV protease overnight in a 3.5 kDa dialysis cassette (Thermo Fisher) floating in a stirred 20 mM Tris-HCl, 100 mM NaCl, pH 8.0 buffer. The TEV reaction was applied to a nickel affinity column to isolate the tag cleaved sample and SEC purification was performed to purify the tagless protein. The final eluate was concentrated to 13 mg/mL and incubated with 1 mM apixaban (Millipore Sigma) in 20 mM HEPES, 50 mM NaCl, pH 7.4 buffer with 5% DMSO prior to crystallization. After screening for crystallization using a Mosquito LCP by STP Labtech, we found that crystals grew in 0.1 M potassium thiocyanate and 30% PEG 2000 MME. Crystals were harvested from a screening tray, cryoprotected with 25% ethylene glycol, and flash frozen in liquid nitrogen. X-ray diffraction was performed at ALS 8.2.1 beamline, data were processed with XDS[56], and the structure was phased by molecular replacement using the designed structure as the search model. The model was refined with Phenix[57] and model building was performed using COOT[58]. The quality of the final model was evaluated using MolProbity[59] (Supplementary Table 3). Data has been deposited in the PDB with accession codes 8VFQ and 8VEZ.

## Multidimensional scaling analysis of structural distances between designed and native NTF2-like structures

We collected 28 natural ketosteroid isomerase (KSI) and ligand complex structures from the PDB (listed in Supplementary Table 5). We assembled a list containing all KSI structures, the computationally designed NTF2 scaffolds from this study (Set 1, 2, and 3), and the characterized binder models. We computed pairwise structural similarities using TM-score[60], selecting the smaller TM-score value when two structure lengths differed. Using (1.0 - TMscore) as the distance metric, we applied multidimensional scaling (mds) using the scikit-learn library. Each NTF2-like structure was visualized in the transformed mds space (Supplementary Fig. 9), and the closest and farthest native KSI structures relative to each characterized binder are reported in Supplementary Table 4.

## Binding assays to test cross-reactivity of the characterized binders

We performed an all by all binding experiment between all targets and representative binders for each target. Binding affinities for the target cortisol (HCY), 17α-hydroxyprogesterone (OHP), warfarin (WRF), and SN-38 were measured by fluorescence polarization. The concentrations of the ligands conjugated to TAMRA were kept constant at 5 nM, 5 nM, 5 nM, and 10 nM for HCY-TAMRA, OHP-TAMRA, WRF-TAMRA, and SN-38-TAMRA, respectively. Fluorescence polarization filter cube (Red1 FP, EX 530/25 nm;EM 590/35 nm, Agilent) was used to measure polarization. BLI was used to determine the $K_D$ values of binders interacting with apixaban (APX-Biotin) and rocuronium (ROC-Biotin). Biotinylated APX and ROC were immobilized to the Streptavidin biosensor and all binders were titrated 3-fold starting from 25uM. Other experimental conditions were kept the same as described above for characterization using FP and BLI. When the $K_D$ value could not be estimated due to weak or no ligand binding, the lowest binding affinity value of 10 mM was assigned to plot Supplementary Fig. 10.

## Computational design of cortisol-induced heterodimers

We employed the hcy129.1_CID-cortisol complex modeled using AlphaFold2 and GALiganddock as our target. This complex features R43I/R95Q/Q128L mutations in addition to hcy129.1, based on the SSM profile of hcy129 to serve as potential hydrophobic protein-protein interaction sites. The AlphaFold2 model of hcy129.1_CID retains the backbone and pocket sidechain structure of hcy129.1. To design minibinders to the NTF2-cortisol interface, we first used PatchDock[30] to find the initial seeding positions of the miniprotein scaffolds against the target interface, and subsequently created Rotamer interaction field (RIF) for both the exposed pocket residues of the NTF2 and the cortisol ligand. The miniprotein scaffold library generated for protein binder design[30] was docked using RIFdock to yield around 5 million total docks. A preliminary 'predictor' design step was used to rank the in silico designs using Rosetta ddG and contact molecular surface in which 1 million docks were selected for the downstream Rosetta design. The interface of the three components; minibinder, hcy129.1_CID, and cortisol, of selected docks were optimized by Rosetta FastDesign[30] with cortisol being explicitly considered by Rosetta. All designs were filtered by contact molecular surface, contact patch, and Rosetta ddG prior to ProteinMPNN sequence redesign where residues within 5 Å of the cortisol ligand were fixed. Finally, we ran Alphafold2 prediction with the initial guess protocol[61] and the designs passing pae_interaction <10 and plDDT_binder > 85 were ordered as a synthetic oligo pool.

## Yeast display and FACS to screen for chemical induced dimerization

Yeast surface display library containing 60k designed minibinders was prepared as previously described[30]. Briefly, the synthetic genes encoding the miniprotein designs were cloned into a pETCON3 yeast surface display vector and the library was transformed into competent EBY100 yeast by electroporation. After the induction of yeast cells in SGCAA medium supplemented with 0.2% glucose, cells were washed with PBSF and incubated with 1 μM purified biotinylated hcy129.1_CID, anti-c-Myc fluorescein isothiocyanate (anti-c-Myc-FITC, Immunology Consultants Laboratory) and streptavidin-phycoerythrin (SAPE, ThermoFisher) in the presence or absence of 1 μM cortisol for 1 hr at room temperature. Cell sorting was performed using a Sony SH800S cell sorter with software version 2.1.5. Three million cells were collected after applying the gate shown in Supplementary Fig. 12A from the library that was incubated with 1uM cortisol. After recovery the collected cells were streaked on C-Trp-Ura agar plates. Colonies were randomly picked and were individually cultured in C-Trp-Ura media, followed by induction in SGCAA media. Yeast display and flow cytometry experiment was done with two conditions for each clone; 1. incubation with 0.2 μM biotinylated hcy129.1_CID, anti-c-Myc-FITC, and SAPE, and 2. Incubation with 0.2 μM biotinylated hcy129.1_CID, anti-c-Myc-FITC, SAPE, and 0.2 μM cortisol. All samples were analyzed with the Invitrogen Attune flow cytometer after washing the cells in PBSF (PBS buffer with 0.1% BSA), and we observed the change in binding signal upon adding cortisol for miniH11 (Supplementary Fig. 12B), which was expressed in *E. coli* for further biochemical characterization (Supplementary Fig. 12C).

## Characterization of cortisol-induced dimerization by SEC

An N-terminal AviTag construct of hcy129.1_CID and C-terminal histag-containing construct of miniH11 were expressed in *E.coli* and purified as described previously (Supplementary Fig. 12C). Briefly, the constructs were expressed in *E. coli* BL21(DE3) and purified by affinity chromatography followed by size-exclusion chromatography. The two protein components of the CID, Nterm-AviTag-hcy129.1_CID and miniH11-HHHHHH, were incubated at 1 μM in the presence or absence of 10 μM cortisol, incubated for -2 h at room temperature, and injected onto an S75 increase 10/300 column with a running buffer of 20 mM HEPES, 50 mM NaCl, pH 7.4. Absorbance was monitored at 280 nm over the course of the elution and resulting elution profiles were overlaid to assess potential cortisol-induced shifts in elution volume.

## Native mass spectrometry

Samples were analyzed by on-line buffer exchange mass spectrometry to evaluate sample purity and the oligomeric states in 200 mM ammonium acetate using a Vanquish ultra-high performance LC system coupled to a Q Exactive ultra-high mass range Orbitrap mass spectrometer (Thermo Fisher Scientific). The buffer exchange columns used included self-packed columns with P6 polyacrylamide gel (Bio-Rad) and the recorded mass spectra were deconvolved and oligomeric assignments were made using UniDec version 4.2+[62].

## Characterization of cortisol-induced dimerization and cortisol sensing with NanoBiT fusions

hcy129.1_CID and miniH11 were genetically fused to SmBiT and LgBiT with empirical GS linkers, respectively, and ordered as eBlocks from IDT. Synthetic genes were cloned into pET29b plasmid and transformed into BL21(DE3) and purified as described above. After purification of the CID sensor components, they were mixed together at 200 nM each and then titrated with variable concentrations of cortisol and incubated for 2 h. Immediately after adding the luciferase substrate diphenylterazine (DTZ) at a final concentration of 25 μM, the luminescent signal was measured on a Neo2 plate reader (Fig. 4H). To estimate the $K_D$ of the dimer by NanoBiT, the miniH11-LgBiT component was kept fixed at 0.1 μM and hcy129.1_CID-SmBiT titrated at variable concentrations (Supplementary Fig. 13B).

## Reporting summary

Further information on research design is available in the Nature Portfolio Reporting Summary linked to this article.

## Data availability

The X-ray crystallographic structure data generated in this study have been deposited in the Protein Data Bank under the accession IDs 8UQF, 8VFQ, and 8VEZ, corresponding to the cortisol-binding protein and the apixaban-binding proteins, respectively. The deep sequencing data have been deposited in the NCBI database with the accession ID PRJNA1356499. Source data are provided with this paper.

## Code availability

The design scripts developed in this study have been deposited on Zenodo [https://doi.org/10.5281/zenodo.17847477].

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

## Acknowledgements

We thank B. Basanta for useful discussions on generating NTF2 scaffolds for binder design. This research was supported by the National Institute on Aging (R01AG063845 to I.G., N.R., and B.C.;R01CA240339 to I.G. and N.R.;R0AI160052 to A.B.); the Open Philanthropy Project Improving Protein Design Fund (to G.R.L., S.J.P., D.T., J.D., A.B., H.N., I.G., and B.C.); the Washington Research Foundation, Innovation Fellows Program (to G.R.L.); a Washington Research Foundation Fellowship (S.J.P.); the NIH Pathway to Independence Award R00EB031913 (A.H.-W.Y.); Department of Defense, Defense Threat Reduction Agency (HDTRA1-21-1-0007 to I.A.;HDTRA1-21-1-0038 to I.G.;HDTRA1-19-1-0003 to S.J.P., G.R.L., and D.T.); the Audacious Project at the Institute for Protein Design (A.K., A.B., H.N., H.H.); Microsoft (D.T., J.D., and I.A.); Howard Hughes Medical Institute (G.R.L., D.B., A.B., I.A., and D.R.L.); the Air Force Office of Scientific Research (FA9550-18-1-0297 to S.P.); Novo Nordisk Foundation (NNF18OC0030446 to C.N.); the National Institute of Allergy and Infectious Diseases (NIAID) (Contract No. 75N93022C00036 and HHSN272201700059C to I.A.); the Defense Advanced Research Projects Agency program Harnessing Enzymatic Activity for Lifesaving Remedies (HEALR) under award HR0011-21-2-0012 (to A.B. and I.G.); National Science Foundation grant CHE-1629214 (A.B.); Spark Therapeutics (I.A.); Bill and Melinda Gates Foundation (#OPP1156262 to A.K. and H.N.); AMGEN (I.G.); NovoNordisk (I.G.); the Nordstrom Barrier Institute for Protein Design Directors Fund (I.G.); Dr. Eric and Ms. Wendy Schmidt, and Schmidt Futures funding from Eric and Wendy Schmidt by recommendation of the Schmidt Futures program (I.G.); R35GM118062 (D.R.L); R01EB031172 (D.R.L.); R01EB027793 (D.R.L.); a Ruth L. Kirchstein National Science Research Award Postdoctoral Fellowship (F32 GM133088 to J.A.M.M.). Crystallographic data were collected at the APS and ALS. Advanced Photon Source (APS) Northeastern Collaborative Access Team beamline 24ID-C, is funded by the National Institute of General Medical Sciences from the National Institutes of Health (P30 GM124165). This research used resources of the Advanced Photon Source, a U.S. Department of Energy (DOE) Office of Science User Facility operated for the DOE Office of Science by Argonne National Laboratory under Contract No. DE-AC02-06CH11357. The Advanced Light Source (ALS) is supported by the Director, Office of Science, Office of 20 Basic Energy Sciences and US Department of Energy under contract number DE-AC02- 05CH11231. The Berkeley Center for Structural Biology is supported in part by the National Institutes of Health (NIH), National Institute of General Medical Sciences. Native mass spectrometry measurements were provided by the NIH-funded Resource for Native Mass Spectrometry Guided Structural Biology at The Ohio State University (NIH P41 GM128577 awarded to Vicki Wysocki).

## Author contributions

D.B. supervised research. G.R.L., S.J.P., and C.N. conceptualized the small-molecule binder design project and performed analysis on binder screening and characterization data. G.R.L. contributed computational metrics for design approach 1, developed the design approach 2 pipeline, experimentally characterized the designs, and participated in LigandMPNN development. S.J.P. redesigned NTF2 scaffolds, formulated design approach 1, and experimentally characterized the designs. C.N. developed the design approach 1 pipeline. A.H-W.Y. conceptualized and characterized the design of chemical-induced dimerizations. S.J.P. and G.R.L. assisted with designing and characterizing cortisol-induced dimers. G.R.L., S.J.P., A.H-W.Y., and D.B. wrote the manuscript. D.T. and I.A. contributed to NTF2 scaffold design. J.D. provided LigandMPNN. J.A.M.M. performed the synthesis of biotinylated ligands under D.R.L.'s supervision. S.J.P., A.K., A.B., and H.N. performed crystallography experiments, data collection, and determined the crystal structure of the cortisol binder. G.R.L., S.J.P., A.K., A.B., E.B., and B.S. performed crystallography experiments, data collection, and determined the crystal structure of the apixaban binder. I.G., D.V., and N.R. helped with yeast library assembly and deep sequencing. H.L.H. assisted with protein purification and BLI experiments. B.C. provided guidance on miniprotein design for the cortisol-induced dimerization system. H.K.H. helped with NGS data analysis.

## Competing interests

G.R.L., S.J.P., C.N., A.H-W.Y., and D.B. are co-inventors on several provisional patent applications submitted by the University of Washington pertaining to the designed protein sequences generated in this work (application numbers: US 63/599,194 and PCT 18/945,961). The remaining authors declare no competing interests.
