## [Transparent Peer Review file · Nature Communications]

Small-molecule binding and sensing with a designed protein family

Corresponding Author: Dr Gyu Rie Lee

Version 0:

Reviewer comments:

Reviewer #3

(Remarks to the Author)

The editor asked me to analyse the other reviewer's report and the authors' response. It is clear that the other reviewer has a different expertise from mine, and I apologize for inaccuracies and incompleteness. I start with the comments from the other reviewer:

* The other reviewer commented that NTF2-fold proteins bind several of the small molecules that the authors targeted in this study, and the authors fully agreed but only in the response letter. I couldn't find these data in the revision, however, and these data are probably the most pertinent to judge the efficacy and generality of the approach. If I understand correctly, the method was successful (many selected binders and high affinity) in cases where NTF2 fold proteins are known to bind similar ligands. This greatly diminishes the claim for generality as we have no demonstration of high-affinity/specificity binders for a molecule that is not already an NTF2 ligand. Therefore, when the authors introduce NTF2 fold proteins and the ligands they tested (in the Introduction), they should explain that NTF2 fold proteins bind chemically similar ligands and provide a range of observed Kds for cognate interactions (and specify these Kds). Because there is extensive background on this that would not be familiar to most protein designers/engineers, the authors should explicitly state which substrates they used are true NTF2-fold targets and what the observed affinities in nature are.

* Regarding warfarin (not a steroid), the authors claim success. Still, the other reviewer points out that the demonstrated affinities are less than an order of magnitude higher than those of warfarin for albumin -- a noncognate protein -- meaning that the designed binder is not much better than a nonspecific one. The comparison with albumin is pertinent as it provides a benchmark for what nonspecific binding would look like. This should be made explicitly in the Results section when the experimental data are provided. Without these points, the results lack essential context. In the Discussion, the authors should return to these points and explain that the ligands they were most successful on are chemically similar to ones to which NTF2 fold proteins have known affinity.

* The other reviewer notes difficulties in interpreting the results regarding rocuronium binding. ITC suggests only 10% occupancy of binding at high ligand concentrations. The reviewer suggests this is due to binder aggregation. The authors partly deal with this criticism by measuring affinity by BLI, but this merely masks the underlying problem that the protein is unlikely to be well-behaved. The stability and homogeneous (non-oligomeric) nature of the designs is critical for successful application and also an important demonstration that the design process is accurate. In the response letter, the authors explain that this problem is likely due to using an older design approach, but they should discuss this in the paper as well.

=====

Although the authors revised their manuscript, some criticisms from my previous review stand as the paper oversells the achievement. I will briefly outline the main criticisms regarding the revision:

* The abstract claims generating "small-molecule binders & sensors for arbitrary ligands remains a grand challenge". But there is sufficient prior work in this field (now cited in the revision) for the authors to speak much more softly. First, previous research has demonstrated small molecule binders with a much more focused effort than done here. Second, the one biosensor demonstrated here is unlikely to be truly useful (more on this below). Third, because the approach is not demonstrated for "arbitrary" ligands -- it was by far most successful on steroid derivatives already known to bind to NTF2 fold proteins (more on this below). In the same vein, while the authors removed the word "general", in various places they claim that the method can be used "to bind and sense a wide range of small molecules", but this phrasing is semantically identical to claiming the method is general.

* With the specificity data provided in the revision, I think the conclusion is that specificity is low. Nevertheless, the authors state it is high because affinity is highest for the cognate-designed ligand. But the usual definition of specificity is the fraction of the target bound by the molecule versus all competitor targets at a given concentration of the molecule. If the affinities for the target molecule and the undesired ones are similar, this fraction would be $\leq 50\%$.

* Related to the above point, in the main text, support for the biosensor comes mainly from Fig. 4H. The authors state that the design shows "sensitivity at low nM range". But at 10nM, luminescence seems to be $<10\%$ of its maximal value. Most importantly, Fig S15 shows that the cortisol response is identical to that of cortisone in that range. Why not integrate this critical figure in the Main figures? I think it is the most relevant for assessing the biosensing potential of the design. As I suggested after my previous reading, the claims about using the designs as biosensors seem inflated.

* the ".binders designed in this study can be readily engineered to serve as sensors for small-molecule of interest". This claims generality, efficacy and ease that are not supported by the data. The paper succeeds on a few molecules that are NTF2 fold protein targets, and the efficacy of the biosensor is low. Furthermore, claiming that the multiple steps of mutagenesis, selection, crystallography and design efforts put into this study make this a demonstration of "ready engineering" is a stretch. Line 329 repeats the claims of efficacy, ease and generality.

To summarize, the de novo protein design field is booming and the labs that contributed to this study are leading this boom. But the potential for applications and the generality of the approach are not proven. The authors concede that more work is required to make gains in affinity and specificity. In addition, they concede that they were successful mostly with molecules for which NTF2 binders are known. The Discussion should state these gaps clearly, providing room for future improvements in the methods.

Minor point:

* It would be helpful if the authors stated the selection criteria and how many exemplars were selected, for instance, lines 156, 170, 177.

(Remarks on code availability)

Reviewer #4

(Remarks to the Author)

Please see attached.

(Remarks on code availability)

I do not see any code that performs ligand docking, a crucial aspect of this paper. Other code for design of proteins (once ligands are docked) looks fine.

Version 1:

Reviewer comments:

Reviewer #3

(Remarks to the Author)

The revision addresses all my concerns and, in my reading, the concerns of the other reviewer that I was tasked with verifying. The revision is an excellent presentation of the significant progress made in this work and additionally highlights areas for future improvement.

(Remarks on code availability)

Dear referees,

We appreciate the thoughtful comments and suggestions. Please find the point by point responses below and the attached revised manuscript. The changes are highlighted in blue.

Reviewer #3 (Remarks to the Author):

The editor asked me to analyse the other reviewer's report and the authors' response. It is clear that the other reviewer has a different expertise from mine, and I apologize for inaccuracies and incompleteness. I start with the comments from the other reviewer:

We thank the reviewer for analyzing the other reviewer's report and our responses as well. We were able to respond to all of the below and also revised the manuscript accordingly.

** The other reviewer commented that NTF2-fold proteins bind several of the small molecules that the authors targeted in this study, and the authors fully agreed but only in the response letter. I couldn't find these data in the revision, however, and these data are probably the most pertinent to judge the efficacy and generality of the approach. If I understand correctly, the method was successful (many selected binders and high affinity) in cases where NTF2 fold proteins are known to bind similar ligands. This greatly diminishes the claim for generality as we have no demonstration of high-affinity/specificity binders for a molecule that is not already an NTF2 ligand. Therefore, when the authors introduce NTF2 fold proteins and the ligands they tested (in the Introduction), they should explain that NTF2 fold proteins bind chemically similar ligands and provide a range of observed Kds for cognate interactions (and specify these Kds). Because there is extensive background on this that would not be familiar to most protein designers/engineers, the authors should explicitly state which substrates they used are true NTF2-fold targets and what the observed affinities in nature are.*

We appreciate the reviewer's comment and we have included analyses on the similarity of the designed NTF2 scaffolds and target ligands to that of the natural NTF2 fold KSI family protein-ligand complex structures found in the PDB. We have updated the manuscript to discuss how these features were related to the design performance in retrospect.

** Regarding warfarin (not a steroid), the authors claim success. Still, the other reviewer points out that the demonstrated affinities are less than an order of magnitude higher than those of warfarin for albumin -- a noncognate protein -- meaning that the designed binder is not much better than a nonspecific one. The comparison with albumin is pertinent as it provides a benchmark for what nonspecific binding would look like. This should be made explicitly in the Results section when the experimental data are provided. Without these points, the results lack*

essential context. In the Discussion, the authors should return to these points and explain that the ligands they were most successful on are chemically similar to ones to which NTF2 fold proteins have known affinity.

We thank the reviewer for this comment. We included an experiment to test binding of albumin to the six targets used in the paper. We updated Supplementary Tables and have added discussions about warfarin binder design, and we have also updated the manuscript to discuss the chemical similarity to the ligands found as complex structures with the KSI protein family (natural NTF2-fold protein) in the PDB.

** The other reviewer notes difficulties in interpreting the results regarding rocuronium binding. ITC suggests only 10% occupancy of binding at high ligand concentrations. The reviewer suggests this is due to binder aggregation. The authors partly deal with this criticism by measuring affinity by BLI, but this merely masks the underlying problem that the protein is unlikely to be well-behaved. The stability and homogeneous (non-oligomeric) nature of the designs is critical for successful application and also an important demonstration that the design process is accurate. In the response letter, the authors explain that this problem is likely due to using an older design approach, but they should discuss this in the paper as well.*

We appreciate this comment and have revised the manuscript to state that the problem of aggregation has still remained after redesign, and we have also modified the text to explain that other design approaches may yield better performing designs.

=====

Although the authors revised their manuscript, some criticisms from my previous review stand as the paper oversells the achievement. I will briefly outline the main criticisms regarding the revision:

** The abstract claims generating "small-molecule binders & sensors for arbitrary ligands remains a grand challenge". But there is sufficient prior work in this field (now cited in the revision) for the authors to speak much more softly. First, previous research has demonstrated small molecule binders with a much more focused effort than done here. Second, the one biosensor demonstrated here is unlikely to be truly useful (more on this below). Third, because the approach is not demonstrated for "arbitrary" ligands -- it was by far most successful on steroid derivatives already known to bind to NTF2 fold proteins (more on this below). In the same vein, while the authors removed the word "general", in various places they claim that the method can be used "to bind and sense a wide range of small molecules", but this phrasing is semantically identical to claiming the method is general.*

We thank the reviewer for the comment. We have revised the abstract to avoid language that suggests broad generality.

** With the specificity data provided in the revision, I think the conclusion is that specificity is low. Nevertheless, the authors state it is high because affinity is highest for the cognate-designed ligand. But the usual definition of specificity is the fraction of the target bound by the molecule*

versus all competitor targets at a given concentration of the molecule. If the affinities for the target molecule and the undesired ones are similar, this fraction would be $\leq 50\%$.

We have reworded this section to more accurately describe the current specificity of the binders and the current limitations.

** Related to the above point, in the main text, support for the biosensor comes mainly from Fig. 4H. The authors state that the design shows "sensitivity at low nM range". But at 10nM, luminescence seems to be $<10\%$ of its maximal value. Most importantly, Fig S15 shows that the cortisol response is identical to that of cortisone in that range. Why not integrate this critical figure in the Main figures? I think it is the most relevant for assessing the biosensing potential of the design. As I suggested after my previous reading, the claims about using the designs as biosensors seem inflated.*

We thank the reviewer for the comment and we have revised the text to explicitly discuss the current limitations of our work particularly regarding target specificity. While we acknowledge that further development is required to address these issues, the primary aim of this work was not to develop a highly specific sensor but to show a design strategy to convert a small-molecule binder to a sensor. Therefore, although we appreciate your thoughts, we retained Fig S15 (now Fig S14, sensor selectivity data figure) in Supplementary figures and kept the main figure 4 to be focused on the conceptual development of the CID system. We have revised the overall manuscript to not overclaim the achievements, and also specifically show the limitations in sensitivity and selectivity.

** the "...binders designed in this study can be readily engineered to serve as sensors for small-molecule of interest". This claims generality, efficacy and ease that are not supported by the data. The paper succeeds on a few molecules that are NTF2 fold protein targets, and the efficacy of the biosensor is low. Furthermore, claiming that the multiple steps of mutagenesis, selection, crystallography and design efforts put into this study make this a demonstration of "ready engineering" is a stretch. Line 329 repeats the claims of efficacy, ease and generality.*

We appreciate the feedback and we modified the manuscript to not overclaim the generality and efficacy of current sensor design. We acknowledge that our method in designing highly sensitive sensors based on binder design is not mature enough, and we have revised the text to state the scope and limitations of this work.

To summarize, the de novo protein design field is booming and the labs that contributed to this study are leading this boom. But the potential for applications and the generality of the approach are not proven. The authors concede that more work is required to make gains in affinity and specificity. In addition, they concede that they were successful mostly with molecules for which NTF2 binders are known. The Discussion should state these gaps clearly, providing room for future improvements in the methods.

We thank the reviewer for the comment, and we have updated the Results and Discussion part to discuss the room for improvements. We have also modified the Conclusion part to discuss related future improvements in the design methods.

Minor point:

** It would be helpful if the authors stated the selection criteria and how many exemplars were selected, for instance, lines 156, 170, 177.*

We switched the orders of the two paragraphs which originally started from lines 156 and 170 for clarity, and added descriptions on how the binder selections for each characterization method were selected in lines 155, 161, and 169.

Reviewer #4 (Remarks on code availability):

I do not see any code that performs ligand docking, a crucial aspect of this paper. Other code for design of proteins (once ligands are docked) looks fine.

Thank you for pointing this out. We will include the code of performing ligand docking during resubmission.

Reviewer #4 (Remarks to the Author):

My review mostly follows the order of the rebuttal letter. If the authors address these concerns, the paper would be ready to publish.

1. The authors write in Para1: "but design methods that can explore the structural diversity of a protein fold while also incorporating precise small-molecule interactions are lacking." As stated, this is not true. The papers they reference used 4-helix bundles built from parametric equations which allowed for systematic variation of the protein backbone to fit molecules like porphyrins, apixaban, and rucaparib.

We thank the reviewer for the comment. We have reworded this section to reframe that the primary advance/advantage here is the nature of the NTF2 fold for small-molecule binder design and its use to design CID, and not any improvement over prior methods in the diversity of protein backbones or the ligands targeted. We updated the manuscript to clarify more details on the design methodology.

2. The 2D map can be misleading. Please make a table of nearest KSI protein with highest Tm value. The binding pockets are the only parts that really need be compared here, since changes in loop length or helix size relative to KSI are not necessarily indicative of different binding sites. Lines 90-94 in the paper need to be quantified. The designs are "distinct" in what way? Because a tsne plot separates them from KSI structures? A better quantitative assessment like nearest Tm score needs to be used (and note which score is used, since for any pair of proteins there are two Tm scores, normalized by one protein's length). I think the Ext Data Fig 9 shows that the hallucinated sets of proteins are exploring a much narrower part of structure space (closer to KSI) than the rosetta method. For those designs to be the most successful perhaps alludes to the native KSI structures being more designable. What do the farthest points from the KSI points in the tsne plot correspond to in Tm score? How close are the backbones of the xtal structures to any of the KSI structures?

We have made Table S4 to report the KSI proteins with the highest and lowest TMscores to each binder characterized for binding (Table S2). The closest KSI structures were mostly bound to Equinelin or Androstanedione, which are both steroid-type ligands, and this may suggest that the designs we computationally made have similar binding pocket structure to the steroid-binding KSI scaffolds. The determined crystal structures of the cortisol binder (hcy129_mpnn) and the apixaban binder (apx1049) have very close structure similarity to the

design model used in this analysis (TMscores 0.93 and 0.97), and we didn't repeat the TMscore calculations to measure the similarity with the KSI structures. Overall, as the reviewer have pointed out, the characterized binders had high structure similarity to the native KSI structures, and this may indicate that although we computationally explored a more diverse space of NTF2-like backbones, the structures that had high similarity to the native KSI structures could be characterized and yielded in putative binders.

We have updated the manuscript to discuss that the NTF2 scaffolds Set 1 and Set 2 (hallucinated) are not distinct but are structurally closer to the KSI structures than Set 3, and also yielded putative binders (removed lines 92-96 and added sentences in line 183-189).

3. Target difference to KSI substrates is not mentioned in the paper. Perhaps a statement that says the affinities of designed binders to targets dropped off as a function of dissimilarity from the KSI substrates. That would capture the essence. It should go in the paper so the reader is aware.

We appreciate this comment and we have made Table S5 which reports the similarity between the target ligands and KSI substrates with known protein-ligand complex structures in the PDB using Tanimoto coefficients. We also added discussion in lines 171-175 stating that the targets with high similarity to the KSI substrates tended to have higher binding affinities to the designs.

4. Any affinity of a molecule that was measured with a linker and biotin/fluorophore attached needs to be measured in a competition experiment with the unlabeled ligand. Or the authors should very prominently display in the main article, main article figures, and supplemental figures, the actual ligand that the binding affinities pertain to. Affinities of unlabeled ligand can differ from labeled ligands by sometimes orders of magnitude (in either direction).

We thank the reviewer and modified the manuscript to prominently describe which form of the target was used to perform the experiments. We updated the text, figure labels (Figure 2), and Table S2 to clarify this information.

5. Re: 7-hydroxy-porgesterone similarity of KSI substrates, the authors need to update the ms to make this clear. The reader should not be expected to know that they differ by a single hydroxyl group.

We appreciate the comment and have included Table S5 reporting the tanimoto coefficients between 17a-hydroxyprogesterone and all KSI substrate ligands.

6. Rocuronium. Despite Fig 2D showing reduced aggregation, Fig 2E still shows a dubious binding isotherm with a 1:10 binding ratio for ROC and a 4uM Kd. The ITC data in the supplement (please put figure names and labels) also still shows a 1:10 binding ratio for ROC22182.1. This discrepancy is not discussed anywhere in the paper, yet it is a major one.

We thank the reviewer for the comment. We tried to improve solubility of roc22182 by reducing aggregation with mutagenesis but as the reviewer pointed out, residual aggregation may have formed and the binding stoichiometry was low. We think this suggests that the binding behavior is still non-ideal, possibly indicating protein or ligand aggregation or cooperativity effects. We have modified the text to discuss in line 197.

7. Warfarin. The authors should perform binding affinity experiments with serum albumin for each target and show in the specificity plot/table. This way the reader will understand how different each design is from an “undesigned” natural promiscuous pocket. In fact, many of the ITC isotherms look to show binding of 2 or more ligands per protein (eg apx1501, wrf7190. Apx1501 shows appreciable higher order oligomer on the SEC trace so perhaps the ligand is binding at an undesigned interface). This needs to be discussed, preferably in the main text of the article. (apx1049 looks 1:1 based on the ITC data, and the xtal structure confirms it is.)

We thank the reviewer and we have included the binding affinity data of bovine serum albumin against the six targets used in this study in Table S6. We now discuss this data in lines 260-273. We also appreciate the comment regarding the non 1:1 binding stoichiometry and oligomer formation of some designs such as apx1501. We have updated the text to discuss this in line 179.

8. Apixaban. It would improve the paper to include some discussion on why the success rates for this target were so low compared to other work. The discussion/conclusion is very terse and could be expanded to address the lack of context to other work and similarities to natural enzymes.

We appreciate the comment and we have included to discuss why the success rate of designing a binder for the target apixaban was low compared to other work in lines 175-182.

9. SN-38. Fig 2E shows FP data which presumably used a fluorophore-tagged molecule. Therefore the reported Kd is not the true Kd and the figure is misleading. The authors should perform competition experiments with the unlabeled molecule to acquire a true Kd. Otherwise they are reporting Kd of the molecule-linker fluorophore, which could be very different. Some of the FP binding titrations still do not appear to saturate (ir807.2 and .3, and ir3009.wt). The reported Kd values are questionable. In general, for FP, the authors should perform competition experiments to obtain a true Kd (although this will not fix the multiphasic issue of some of the binders, and a Kd with the linked fluorophore is still needed to fit a true Kd). Also see Supp Table 2 for reported FP values. These are not Kds of the actual ligands, but of the modified ligands, so update the table/figures or perform the competition experiments.

We thank the reviewer for the comment and updated the corresponding text, figure labels, and Table S2 to clarify the type of ligands used for each binding assay. For example, we have clarified that the ligand used to test binding by FP was modified with a fluorophore (SN-38-TAMRA) in the manuscript.

10. *Strictly monomers. The SEC plots are hard to interpret. These proteins are roughly the same size and shape so they would be expected to sediment at the roughly the same rate. Some seem to elute at 14 mL and some closer to 15-16 mL. It would be best for the authors to label the SEC curves with the expected MW of the protein based on a standard curve, along with the actual MW of the protein. Perhaps the authors should elaborate in the discussion about how their desired goal of monomeric proteins seem to be by and large at odds with what they observed. They were able to obtain some monomeric binders, but many of their binders were oligomers, which makes the claim of the number of “successful” designs a bit dubious here other than for the most characterized, monomeric (and experimentally optimized) binders.*

We appreciate the comment and we have modified Figure 2 by adding labels with expected MW of the proteins and the expected elution volume with arrows. The standard curve can be found in Supplementary Information. We agree that some of the binders exhibited SEC profiles indicating species of larger apparent size than expected for a monomer, which may reflect oligomerization or altered folding states. We removed the word ‘successful’ and modified the manuscript to discuss that within the characterized binders, some exhibited monomeric SEC profiles and they tended to result in higher binding affinity in lines 175-181.

Fig 3. Please show density of the interacting sidechains.

We have modified Figure 3 and now it shows the density of interacting side chains.

The paper is terse on insight. More discussion about what they learned would be appreciated by the wider community.

We thank the reviewer for this comment and we have added more discussion on the current limitations particularly focused on target specificity and sensitivity, and low success rate that can potentially be improved with more method development. Discussions have been added in the Results and Discussion and Conclusion sections.

Wrf21208, wrf5284, wrf7190 are not in the SEC trace fig Ext data fig 5. Are they aggregating? Wrf5461 is clearly an aggregate by SEC yet a Kd of 10 uM is still reported. How did the authors model the aggregate during the fit? If they used a 1:1 binding model, this would result in high error.

We appreciate the reviewer for this comment. We agree that the binding affinities should be reported for well-behaving proteins. We updated the manuscript (line 164) and Table S2 to exclude data from binders with non-ideal SEC traces. This led to reporting two WRF binders with the binding affinities characterized with ITC.

Line 230: “We performed binding assays for all target and binder pairs to test crossreactivity” The authors only chose 6 proteins, not “all”. Please report these affinities with real numbers and error bars, not just colors on a heat map.

Thank you for pointing this out. We updated our text to clarify the number of binders used for this experiment and we have included Table S6 to report the binding affinities (K_D (μ M)) of each of the six representative binders measured against six targets.

Line 233: "Overall, the designed binders bound to their target small molecules specifically". This is not what the data shows. Please change this statement. "cross-talk with structurally similar ligands" is also a bit misleading. If there is this much "cross-talk" between this limited set of ligands, these designs might indeed be quite promiscuous once the set is even marginally expanded.

We agree with the comment and we have modified the manuscript (lines 260-273) and included Table S6.

I cannot find any provided code or detailed methods for docking the ligands into the proteins. This is an important part of the paper but appears to be completely missing in description.

Thank you for the comment. We will provide the scripts to perform Rfdock which we used for ligand docking during resubmission. We have also updated the manuscript to add more details on the docking method in line 390 and 407 in the Methods section.

Designs look to use only ser/thr/his/trp/tyr for hbonds?

We didn't explicitly restrict the design method to use the amino acids ser, thr, his, trp or tyr for hydrogen bonding. We have allowed our sequence design methods RosettaFastDesign or LigandMPNN to use all amino acid types, including other charged amino acids such as asp, glu, or lys, but in retrospect, the selected candidate binder designs predominantly featured ser/thr/his/trp or tyr. This preference likely arises because incorporating charged residues such as asp/glu/lys/arg at designed protein-ligand interfaces will require a precise formation of charge interaction networks, which can otherwise destabilize the binding site, and the designed binding sites were especially situated near the core of the NTF2 fold proteins utilized in this work. The designs which included other charged residues (asp,glu,lys, arg, ...) were mostly filtered out computationally due this aspect. We have updated the manuscript to add details on or docking methods as described above.

Ext data fig 5 looks messy. Lots of oligomers. The authors should comment on the correlation between their selected binders and aggregated proteins.

We included the SEC profiles of all 33 purified proteins of the designs selected from yeast display binding screening. 23 of these that had distinct peaks that could be separated were further characterized for binding. We have modified the manuscript in lines 175-182 to discuss the correlation between oligomerization and binding.

Ext data fig 8 shows representative density but not of protein-ligand contacts...

We have modified Figure 3 to represent the density of protein sidechains interacting with the ligand.

Ext data fig 9, tsne or umap is likely exaggerating the distances here. Please show the raw Tm scores of each design closest to a KSI structure. Put the numbers in a table. I don't see any details in the ms for how the map was made.

We have included Table S4 to report the structurally closest (highest TMscore) KSI structure for each design characterized for binding. Based on the results, we agree with the reviewer's comment that the multi dimensional scaling (mds) plot looks exaggerated than the actual structural differences. We appreciate this comment and we modified the text (removed line 92 and added lines 171-177) to state that in retrospect, the characterized binders and the scaffolds used to generate the binders were structurally similar to the native KSI structures. We have also updated the manuscript to add a description on how the map was made in Methods (line 620).

Ext data fig 10. Which binders are these? I assume they have names that can be tracked back to design models. The figure is ambiguous. Also, the log scale makes it difficult to know the Kd, so please put the numbers explicitly in the plot or in a table. Please also include the Kd of each ligand with albumin for reference.

We appreciate the comment and have added Table S6. to report the binding affinities (K_D (μM)) of each of the six representative binders measured against all six targets that were used to make Extended data Fig. 10. We also updated the figure legend.

Ext data fig 12. C shows oligomer (two peaks). What is MW state of D and what does it run at?

Extended data Figure 12.C shows the SEC plot of hcy129.1_CID, which is a variant of the monomer cortisol binder designed to dimerize with another miniprotein (Ext data Fig 12. D). Therefore, we only collected the peak that elutes at approximately 15 mL to perform the following experiments. We would expect oligomeric states (dimer/trimers) of the hcy129.1_CID construct to elute at approximately 12 mL ('MW state of D') and thus did not use this peak in downstream sensing experiments.

Ext data fig 14. A. Why would the ternary complex be expected to fly together in mass spec? They are not covalently attached, nor do they bind very tightly. B. The curve does not appear to level off, so the Kd is likely a lower bound.

The type of mass spectrometry performed was native mass spectrometry (<https://pubs.acs.org/doi/10.1021/acs.chemrev.1c00212>), which maintains complexes unlike standard mass spec, so we expect to see the mass for the dominant species present in solution. We specifically chose the method to observe the formation of the complex.

We have changed the label in Extended data Figure 14 (**Now Extended data Fig 13**) to read as > 5 μM instead of approximately 5 μM to better represent the binding affinity of this complex and this change is also applied in the main text.

We decided to exclude **Extended data Fig 13. “SEC-based screen of a solution of hcy129.1_CID (1 μM) and indicated miniproteins (1 μM) in the presence (peach) or absence (black) of cortisol (10 μM).”, as we mainly used yeast display and flow cytometry to identify the miniprotein (miniH11) that showed increasing binding to hcy129.1_CID with the presence of cortisol (Extended data Fig. 12, and we have updated the manuscript to add more details on the Method (line 648~658)).

Ext data fig 15. Figure legend should have reaction conditions, concentrations.

We have added the reaction conditions and protein concentrations to the figure legend of ext data fig 15 (Now **Extended data Fig 14**).

Fig. 2. Which proteins are these? Are all of these from computational designs or some from experimental optimization? (At least iri protein is a redesign to be more soluble.) I assume they have names that can be tracked throughout the ms? Iri807 design is not in its binding isotherm figure in supplement. Please put figure legends with figure labels on the supplemental figures. Fig 2B says “Design /// AlphaFold”. Did some of these designs come from combinatorial mutagenesis (It would be easier to know which if they were appropriately labeled.)? The figure should make clear which designs were from which round of computation and which designs include experimental optimization of the sequence.

We appreciate the comment and modified Figure 2 to clearly specify which round of computation the reported designs are from. We have added labels of the designs in panel B and the design names can be tracked throughout the text and Table S2. Except for the rocuronium binder (roc22182.1), which was the design redesigned with mutagenesis to improve solubility, we reported data of the initially designed binders (first round without optimization) for the other targets. Iri807 design was characterized by FP and we modified the labels to state this.

My review mostly follows the order of the rebuttal letter. If the authors address these concerns, the paper would be ready to publish.

1. The authors write in Para1: “but design methods that can explore the structural diversity of a protein fold while also incorporating precise small-molecule interactions are lacking.” As stated, this is not true. The papers they reference used 4-helix bundles built from parametric equations which allowed for systematic variation of the protein backbone to fit molecules like porphyrins, apixaban, and rucaparib.
2. The 2D map can be misleading. Please make a table of nearest KSI protein with highest T_m value. The binding pockets are the only parts that really need be compared here, since changes in loop length or helix size relative to KSI are not necessarily indicative of different binding sites. Lines 90-94 in the paper need to be quantified. The designs are “distinct” in what way? Because a tsne plot separates them from KSI structures? A better quantitative assessment like nearest T_m score needs to be used (and note which score is used, since for any pair of proteins there are two T_m scores, normalized by one protein’s length). I think the Ext Data Fig 9 shows that the hallucinated sets of proteins are exploring a much narrower part of structure space (closer to KSI) than the rosetta method. For those designs to be the most successful perhaps alludes to the native KSI structures being more designable. What do the farthest points from the KSI points in the tsne plot correspond to in T_m score? How close are the backbones of the xtal structures to any of the KSI structures?
3. Target difference to KSI substrates is not mentioned in the paper. Perhaps a statement that says the affinities of designed binders to targets dropped off as a function of dissimilarity from the KSI substrates. That would capture the essence. It should go in the paper so the reader is aware.
4. Any affinity of a molecule that was measured with a linker and biotin/fluorophore attached needs to be measured in a competition experiment with the unlabeled ligand. Or the authors should very prominently display in the main article, main article figures, and supplemental figures, the *actual* ligand that the binding affinities pertain to. Affinities of unlabeled ligand can differ from labeled ligands by sometimes orders of magnitude (in either direction).
5. Re: 7-hydroxy-porgesterone similarity of KSI substrates, the authors need to update the ms to make this clear. The reader should not be expected to know that they differ by a single hydroxyl group.
6. Rocuronium. Despite Fig 2D showing reduced aggregation, Fig 2E still shows a dubious binding isotherm with a 1:10 binding ratio for ROC and a 4uM K_d. The ITC data in the supplement (please put figure names and labels) also still shows a 1:10 binding ratio for ROC22182.1. This discrepancy is not discussed anywhere in the paper, yet it is a major one.
7. Warfarin. The authors should perform binding affinity experiments with serum albumin for each target and show in the specificity plot/table. This way the reader

will understand how different each design is from an “undesigned” natural promiscuous pocket. In fact, many of the ITC isotherms look to show binding of 2 or more ligands per protein (eg apx1501, wrf7190. Apx1501 shows appreciable higher order oligomer on the SEC trace so perhaps the ligand is binding at an undesigned interface). This needs to be discussed, preferably in the main text of the article. (apx1049 looks 1:1 based on the ITC data, and the xtal structure confirms it is.)

8. Apixaban. It would improve the paper to include some discussion on why the success rates for this target were so low compared to other work. The discussion/conclusion is very terse and could be expanded to address the lack of context to other work and similarities to natural enzymes.
9. SN-38. Fig 2E shows FP data which presumably used a fluorophore-tagged molecule. Therefore the reported Kd is not the true Kd and the figure is misleading. The authors should perform competition experiments with the unlabeled molecule to acquire a true Kd. Otherwise they are reporting Kd of the molecule-linker-fluorophore, which could be very different. Some of the FP binding titrations still do not appear to saturate (ir807.2 and .3, and ir3009.wt). The reported Kd values are questionable. In general, for FP, the authors should perform competition experiments to obtain a true Kd (although this will not fix the multiphasic issue of some of the binders, and a Kd with the linked fluorophore is still needed to fit a true Kd). Also see Supp Table 2 for reported FP values. These are not Kds of the actual ligands, but of the modified ligands, so update the table/figures or perform the competition experiments.
10. Strictly monomers. The SEC plots are hard to interpret. These proteins are roughly the same size and shape so they would be expected to sediment at the roughly the same rate. Some seem to elute at 14 mL and some closer to 15-16 mL. It would be best for the authors to label the SEC curves with the expected MW of the protein based on a standard curve, along with the actual MW of the protein. Perhaps the authors should elaborate in the discussion about how their desired goal of monomeric proteins seem to be by and large at odds with what they observed. They were able to obtain some monomeric binders, but many of their binders were oligomers, which makes the claim of the number of “successful” designs a bit dubious here other than for the most characterized, monomeric (and experimentally optimized) binders.

Fig 3. Please show density of the interacting sidechains.

The paper is terse on insight. More discussion about what they learned would be appreciated by the wider community.

Wrf21208, wrf5284, wrf7190 are not in the SEC trace fig Ext data fig 5. Are they aggregating? Wrf5461 is clearly an aggregate by SEC yet a Kd of 10 uM is still reported. How did the authors model the aggregate during the fit? If they used a 1:1 binding model, this would result in high error.

Line 230: “We performed binding assays for all target and binder pairs to test cross-reactivity” The authors only chose 6 proteins, not “all”. Please report these affinities with real numbers and error bars, not just colors on a heat map.

Line 233: “Overall, the designed binders bound to their target small molecules specifically”. This is not what the data shows. Please change this statement. “cross-talk with structurally similar ligands” is also a bit misleading. If there is this much “cross-talk” between this limited set of ligands, these designs might indeed be quite promiscuous once the set is even marginally expanded.

I cannot find any provided code or detailed methods for docking the ligands into the proteins. This is an important part of the paper but appears to be completely missing in description.

Designs look to use only ser/thr/his/trp/tyr for hbonds?

Ext data fig 5 looks messy. Lots of oligomers. The authors should comment on the correlation between their selected binders and aggregated proteins.

Ext data fig 8 shows representative density but not of protein-ligand contacts...

Ext data fig 9, tsne or umap is likely exaggerating the distances here. Please show the raw Tm scores of each design closest to a KSI structure. Put the numbers in a table. I don't see any details in the ms for how the map was made.

Ext data fig 10. Which binders are these? I assume they have names that can be tracked back to design models. The figure is ambiguous. Also, the log scale makes it difficult to know the Kd, so please put the numbers explicitly in the plot or in a table. Please also include the Kd of each ligand with albumin for reference.

Ext data fig 12. C shows oligomer (two peaks). What is MW state of D and what does it run at?

Ext data fig 14. A. Why would the ternary complex be expected to fly together in mass spec? They are not covalently attached, nor do they bind very tightly. B. The curve does not appear to level off, so the Kd is likely a lower bound.

Ext data fig 15. Figure legend should have reaction conditions, concentrations.

Fig. 2. Which proteins are these? Are all of these from computational designs or some from experimental optimization? (At least iri protein is a redesign to be more soluble.) I assume they have names that can be tracked throughout the ms? Iri807 design is not in its binding isotherm figure in supplement. Please put figure legends with figure labels on the supplemental figures. Fig 2B says “Design /// AlphaFold”. Did some of these designs come

from combinatorial mutagenesis (It would be easier to know which if they were appropriately labeled.)? The figure should make clear which designs were from which round of computation and which designs include experimental optimization of the sequence.